# Development and preliminary validation of a land surface image assimilation system based on the common land model

Wangbin Shen[1], Zhaohui Lin[2], Zhengkun Qin[1], Juan Li[3]

[1]Center of Data Assimilation for Research and Application, Nanjing University of Information Science and Technology, Nanjing, 210044, China
[2]International Center for Climate and Environment Sciences, Institute of Atmospheric Physics, Chinese Academy of Sciences, Beijing, 100029, China
[3]CMA Earth System Modeling And Prediction Center, China Meteorological Administration, Beijing, 100086, China

*Correspondence to*: Zhaohui Lin (lzh@mail.iap.ac.cn); Zhengkun Qin (001771@nuist.edu.cn)

**Abstract.** Data assimilation is an essential approach to improve the predictions of land surface models. Due to the characteristics of single-column models, assimilation of land surface information has mostly focused on improving the assimilation of single-point variables. However, land surface variables affect short-term climate more through large-scale

anomalous forcing, so it is indispensable to pay attention to the accuracy of the anomalous spatial structure of land surface variables. In this study, a land surface image assimilation system capable of optimizing the spatial structure of the background field is constructed by introducing the curvelet analysis method and taking the similarity of image structure as a weak constraint. The ERA5-Land soil moisture reanalysis data are used as ideal observation for the preliminary effectiveness validation of the image assimilation system. The results show that the new image assimilation system is able to well absorb

the spatial structure information of the observed data and has a remarkable ability to adjust the spatial structure of soil moisture in the land model. The spatial correlation coefficient between model surface soil moisture and observation has increased from 0.39 to about 0.67 after assimilation. By assimilating the surface soil moisture data and combining with the model physical processes, the image assimilation system can also gradually improve the spatial structure of soil moisture content at a depth of 7–28 cm, with the spatial correlation coefficient between model soil moisture and observation increased

from 0.35 to about 0.57. The forecast results show that the positive assimilation effect could be maintained for more than 30 days. The results of this study adequately demonstrate the application potential of image assimilation system in the short-term climate prediction.

## 1 Introduction

Soil moisture not only affects surface processes such as dust (Lei et al., 2005), but also progressively influences climate change by altering surface albedo, heat capacity, and sensible heat and latent heat transported to the atmosphere (Lin et al., 2001; Li et al., 2019; Zhou et al., 2020a). Soil moisture changes slowly relative to the atmospheric variables, that is, the soil moisture has long-term memory. The initial soil moisture anomaly in the sub-seasonal to seasonal forecasting system can be transferred into the forecast, and thus it is an important source of sub-seasonal climate predictability (Koster et al., 2020). Accurate initial land surface conditions can remarkably improve the accuracy of climate and hydrological projections in short-term climate prediction, especially in fully coupled numerical models (Zhan and Lin, 2011;Wang and Cui, 2018; Zheng et al., 2018; Crow et al., 2020; Reichle et al., 2021; Cui and Wang, 2022).

Based on the comprehensive consideration of observation and model errors, the land surface data assimilation method effectively integrates the model background field and various types of observational data with different spatio-temporal distributions and error characteristics, so as to obtain the optimal initial conditions of soil moisture (Li et al., 2020a; Naz et al., 2020). Research on land surface data assimilation methods has gained the attention of meteorologists around the world. Initially, the European Centre for Medium-Range Weather Forecasts (ECMWF) used the nudging method to adjust land surface variables based on the relationship between the forecast errors of atmospheric variables (e.g. specific humidity at near-surface level) and soil moisture errors (Douville et al., 2000). In order to make the assimilated analysis fields better coordinated with other variables of the model, Mahfouf (1991) proposed the optimal interpolation (OI) scheme to assimilate near-surface temperature and humidity observations. The four-dimensional variational data assimilation (VDA) method has also been applied in the study of land surface data assimilation (Reichle et al., 2001). However, for the complex land surface models with strong nonlinearity, it is difficult to compile adjoint models (Dunne and Entekhabi, 2005). Therefore, the VDA is barely used in land surface assimilation. The Kalman filter-like assimilation with no adjoint models is more widely used for land surface data assimilation (Tian et al., 2008; Jin and Li, 2009; Jia et al., 2010; Shi et al., 2011; Sabater et al., 2019; Tangdamrongsub et al., 2020).

Soil moisture assimilation has been conducted at different spatial scales using a range of methods such as the VDA and Kalman filtering (Gruber et al., 2018; De Santis et al., 2021; Sabater et al., 2019). Stable assimilation also remarkably improves soil moisture prediction (Khaki et al., 2020). However, in order to accommodate the features of the single-column land surface model, the current land surface assimilation system is a single-column assimilation system, which neglects the spatial continuity of soil variables. On the timescale of short-term climate change, soil moisture is commonly responsible for the abnormal changes in short-term climate through long-term and large-scale anomalous forcing (Lin et al., 2008; Zhong et al., 2020; Dirmeyer et al., 2021). The significant influence of soil moisture on local precipitation has been extensively studied, revealing regional variations in the underlying mechanisms (Douville et al., 2001; Cioni and Hohenegger, 2017). Additionally, soil moisture can also trigger the atmospheric teleconnection wave trains or inducing large-scale circulation

anomalies through impacting surface energy balance, which subsequently manifest as non-local and large-scale climate effects (Gao et al., 2020). Therefore, improving the accuracy of the anomalous spatial structure of land surface variables, which serve as the lower boundary conditions of numerical models, will help to better predict short-term climate change caused by soil moisture anomalies.

Ideally, a single-column assimilation system would also be able to reproduce the correct spatial-structure features of soil moisture anomalies, if the assimilation can obtain the closest result to the true value at each column.  Due to the non-uniform spatial distribution of precipitation, as well as the heterogeneous spatial distribution of soil properties, land cover types and topographic elevations, there are significant variations in the spatial distribution of soil moisture (Tian et al., 2021). The estimation of soil moisture by the land surface model is adversely impacted by the uncertainties in atmospheric forcing,

model dynamics and parameterization, leading to significant spatial variations in the accuracy of simulated surface variables (Li, 2013; Li et al., 2020b). Furthermore, there are regional differences in the accuracy of the estimation of the observation error and the background error resulting from the single column assimilation, which ultimately contribute to the discontinuity of the abnormal spatial structure in the analyzed soil moisture field. The estimation of single-point observation error and background error through statistical methods is characterized by significant uncertainty, while point-by-point

assimilation methods have limitations in capturing spatial information from neighboring pixels. In addition, the bias correction is commonly employed to rectify the discrepancy between model simulations and observations prior to assimilation. The prevailing assimilation system primarily addresses the bias by incorporating scale adjustments into the model simulation based on observed data. The spatial distribution structure information, however, is compromised as a result of rescaling (Zhou et al., 2019). Zhou et al. (2020b) also pointed out that most current soil moisture assimilation methods

eliminate the systematic biases between observation and simulation by applying the pixel-by-pixel scale transformation. This treatment discards the crucial spatial information contained in the observation, and affects the application of soil moisture in numerical weather prediction, flood forecasting and drought monitoring. Therefore, while data assimilation improves the accuracy of single-point soil variables, appropriately adjusting the spatial structure of soil analysis variables is a critical development direction for land surface assimilation systems. The uneven spatial distribution of precipitation and the

heterogeneousness of soil properties, land cover types and topography would result in significant spatial variations in the characteristics of soil moisture (Tian et al., 2021). The effectiveness of estimating soil moisture using observational data is limited due to significant spatial heterogeneity. Therefore, a lot of studies strive to incorporate spatial structure information from soil moisture observation to land data assimilation, to enhance the accuracy of spatial pattern of soil moisture to the greatest extent possible (Pauwels et al., 2001; Han et al., 2012; Zhou et al., 2019). Enhancing soil moisture levels is of

utmost importance; however, it is equally crucial to acquire a more precise comprehension of the spatial distribution of soil moisture for effective management strategies, particularly in key regions like the Qinghai-Tibet Plateau where land-air interactions are significant and there are large spatial variations of soil moisture.

With the continuous development of meteorological observation techniques, more and more meteorological information can be displayed in the form of images with fine spatio-temporal resolutions, and their continuous dynamic changes

**Formatted:** Font color: Auto

**Deleted:** Therefore, a lot of studies strive to acquire the precise spatial structural information of soil moisture to the greatest extent possible. Pauwels et al. (2001) employed the nudging technique to incorporate spatial structure information derived from remote sensing soil moisture observations, and obtained enhanced predictions of runoff. Han et al. (2012) examined the constraints of introducing the horizontal correlation features of satellite soil moisture observation data during land surface data assimilation. The findings demonstrated that incorporating surrounding observations and spatial horizontal correlation structure information may improve the analysis field of soil moisture in uncovered grids. The regional soil water assimilation scheme developed by Zhou et al. (2019) incorporates an empirical approach and accounts for spatial variability, resulting in significantly improved accuracy of soil moisture simulation in both temporal and spatial dimensions. The findings of these studies suggest that enhancing soil moisture levels is of utmost importance; however, it is equally crucial to acquire a more precise comprehension of the spatial distribution of soil moisture for effective management strategies, particularly in key regions like the Qinghai-Tibet Plateau where land-air interactions are significant and there are large spatial variations of soil moisture.

**Formatted:** Font color: Text 1

generally allow us to better understand the observed variables. However, the huge amount of satellite observation images for the earth system are not sufficiently utilized in the current data assimilation system (Vidard et al., 2008). Stroud et al. (2009) developed several assimilation schemes that combined the images obtained from the Sea-viewing Wide Field Sensor with a two-dimensional sediment transport model of Lake Michigan, which considerably improved the predictions of sediment concentrations in southern Lake Michigan. In order to quantitatively assimilate the structural information contained in images or image sequences into the numerical model, Le Dimet et al. (2015) extracted the key structural observation information in the images by using curvelet transformation as the observation operator, and improved the prediction results of the shallow water equation model through the VDA approach. Titaud et al. (2010) also found that the direct VDA of image sequences is able to reconstruct initial vortices with highly correct positions, sizes and profiles by using the curvelet transform as the observation operator. Currently, direct assimilation of image sequences is primarily used to predict the evolution of geophysical fluids. If the structural information in the observed images can be introduced into the land surface data assimilation system as observations, the accuracy of the spatio-temporal distribution structure of soil moisture in the model can be targetedly improved. The purpose of this study is to construct a land surface image assimilation system based on the theoretical frame of VDA, so as to realize the direct adjustment of the spatial structure of land surface variables and improve the accuracy of the initial soil moisture values, by integrating the observation information of image sequences and the priori knowledge from numerical models. In this study, an attempt is made to test the effectiveness of the image assimilation module in improving the spatial structure of soil moisture at the land surface in a VDA framework, and the related research methodology can be implemented in the alternative assimilation framework as well.

The study area selected in this research is mainly East Asia, encompassing the alpine regions of Siberia, the vegetative regions of eastern China, as well as the Qinghai-Tibet Plateau and desert regions of western China. The estimation of observation error and model error becomes more challenging in the Tibetan Plateau region, particularly for single point assimilation. Including the plateau region can effectively showcase the advantages of image assimilation method. The paper is organized as follows. Section 2 mainly introduces how to select appropriate image observation operators and establish an image assimilation system under the VDA framework. Then, the error characteristics of the image observation operators are systematically analyzed. The land surface model and the observational data used in the assimilation system are also briefly described in section 2. Section 3 presents the experimental designs, and analyzes the error characteristics of background field and observation data in detail. The results of the idealized experiments are shown in section 4 to verify the effectiveness of the image assimilation system in improving the predictions of the land surface model. Section 5 gives the summary and discussion.

## 2 Construction of image assimilation system

### 2.1 Land surface model

The common land model (CoLM) developed by Dai et al. (2003) was selected in this study. By considering biophysical, biochemical, ecological and hydrological processes, this model well describes the transfer processes of energy, water and carbon dioxide among soil, vegetation, snow cover and atmosphere, allowing the simulation of soil temperature, soil moisture, runoff, heat flux, and other variables. In recent years, the CoLM has incorporated additional physical processes such as glaciers, lakes, wetlands and dynamic vegetation. It has also been successfully implemented in several global atmospheric models (Yuan and Liang, 2011; Ji et al., 2014; Zhang et al., 2020; Yuan and Wei, 2022).

The surface spatial heterogeneity of the CoLM is manifested as a nested sub-grid hierarchy, with the grid units consisting of multiple land units and plant function types (PFTs). The bio-geophysical processes of the CoLM are simulated on a single soil-vegetation-snow column, and each sub-grid has its own predictor variables. Grid-averaged atmospheric forcing is used to force all sub-grid cells within a grid cell. The model used in this study has a horizontal resolution of about 1.4° × 1.4°. There are ten unevenly spaced soil layers and a maximum of five snow layers in the vertical direction.

Soil moisture and its vertical transport is governed by infiltration, runoff, gradient diffusion, gravity, and root extraction by canopy transpiration. Only the vertical transport of soil water is considered in the CoLM model. The water in the soil will percolate through the soil pores due to the combined effects of gravity and capillary forces. According to the principle of mass conservation, the vertical movement of soil water can be mathematically described by the Richards equation.

$$\frac{\partial \theta}{\partial t} = -\frac{\partial q}{\partial z} - E - R_{fm} ,\qquad(1)$$

where $\theta$ is the volumetric water content of the soil (unit: m³·m⁻³), $q$ the soil water flux calculated by the Darcy theorem, $E$ the rate of evaporation (unit: mm·s$^{-1}$), $R_{fm}$ the rate of thawing or freezing, and $z$ the vertical distance from the soil layer to the ground ( $q$ and $z$ is downward-positive).

Atmospheric forcing conditions provide constraints on land-surface models. The atmospheric forcing dataset used to drive the CoLM in this study includes the downward short-wave solar radiation at surface, downward long-wave radiation, near-surface air temperature, specific humidity, precipitation rate, surface atmospheric pressure, U-component wind speed, and V-component wind speed. It has a temporal resolution of three hours (at 0000 UTC, 0300 UTC, 0600 UTC, etc.) and the spatial resolution is T62 (about 1.875°) (Qian et al., 2006). The forcing dataset was derived through combining observation-based analyses of monthly precipitation and surface air temperature with intramonthly variations from the National Centers for Environmental Prediction-National Center for Atmospheric Research (NCEP-NCAR) reanalysis. To correct the spurious long-term changes and biases in the NCEP-NCAR reanalysis precipitation, surface air temperature, and solar radiation fields,

Qian et al. (2006) combined the intramonthly variations from the NCEP-NCAR 6 hourly reanalysis with monthly time series derived from station records of temperature and precipitation. It is shown that the CLM 3.0 reproduces many aspects of the long-term mean, annual cycle, interannual and decadal variations when it was forced by this dataset.

In this study, the CoLM is run in the offline mode cyclicly driven by the observation-based forcing data from 1948 to 2020 for 360 years. The water content of the deepest layer changes extremely slowly over the last 50 years, and the model can be considered to be in equilibrium.

## 2.2 Framework of image assimilation system based on variational data assimilation

This study is based on the framework of three-dimensional VDA (3D-VDA). The main principle of 3D-VDA is to simplify data assimilation to a quadratic functional minimization problem, which characterizes the deviations between analysis and observational fields as well as between analysis and background fields.

Assuming that $x$ denotes the vector of analysis variables, $x_b$ denotes the background field, and $x_a$ denotes the analysis field, then the variation of $x$ with time can be expressed as:

$$\begin{cases} x(t) = M\big(x(t_0)\big) + \eta(t) \\ x(t_0) = U \end{cases}, \tag{2}$$

where $M$ denotes the numerical prediction model, and $t$ and $t_0$ represents the prediction time and start time of the model, respectively. $\eta(t)$ is the model error at moment $t$, $U \in \mathbb{R}$ denotes the initial conditions of the model, and $\mathbb{R}$ represents the space in which the state variables are located.

If a direct or indirect observation vector of length $L$ is represented by $y^o \in \mathbb{Q}$ and the observation space is represented by $\mathbb{Q}$, then the relationship between observations and state variables can be expressed as follows :

$$y^o(t) = H\big(x(t)\big) + \mathcal{E}, \tag{3}$$

where $H: \mathbb{R} \to \mathbb{Q}$ is the observation operator that represents a mapping from the model space to the observation space. The observation operator is simplified to a simple interpolation operator when $y^o$ and $x$ are the same type of physical variable. If the two have different physical properties, the observation operator is a mapping operator with some complex structure that transforms the model space into the observation space. $\mathcal{E}$ represents observation error. The goal of variational assimilation is to determine the model state at time $t_0$ , so that the sum of the deviation of state variable from background field and the deviation of simulated observation based on model variable from actual observation, is minimized under the premise of additional constraints, that is, to find an analysis field $x_a(t_0)$ which minimizes the following quadratic objective function $J$.

$$J(x) = \frac{1}{2}[x - x_b]^T B^{-1}[x - x_b] + \frac{1}{2}[H(x) - y^o]^T (O + F)^{-1}[H(x) - y^o] , \tag{4}$$

Where $B$, $O$ and $F$ are respectively the error covariance matrixes of background field, observation data and observation operator, which are known as prior information. $B^{-1}$ are the inverse of the background error covariance matrix with order $N \times N$, and $N$ is the freedom degree of the analysis field. $(O + F)^{-1}$ is the inverse of the observation error covariance matrix with order $L \times L$.

     During minimizing the above objective function, the optimal analytical variable is $x = x_a$ when $\nabla_x J(x_a) = 0$. It

represents the optimal estimate of the true atmospheric state under given background fields, observations, and their error information.

     Images are generally characterized by the features of observation variables, such as geometry and distribution. From a "mathematical" point of view, images are usually considered to be real-valued functions of consecutive real variables, so they can be processed by using mathematical tools. In this case, the "numerical image" is a discrete version of the final

processed mathematical image (Le Dimet et al., 2014).

     The so-called image assimilation refers to the introduction of a weak constraint on the similarity between the structure of the observed and simulated images in the VDA, so that the image observations are used together with the conventional observations to compute the optimal analysis variables. Thus, the cost function for the 3D-VDA can be written in the following form:

$$J(x) = J_B + J_O + J_I = \underbrace{\frac{1}{2}(x_o - x_b)^T B^{-1}(x_o - x_b)}_{conventional\ cost\ of\ J_B} + \underbrace{\frac{1}{2}(H(x_o) - y^o)^T (O + F)^{-1}(H(x_o) - y_o)}_{conventional\ cost\ of\ J_O} +$$

$$\underbrace{\frac{1}{2}\left(H_{F \to s}(f_o) - H_{\mathbb{R} \to s}(x_b)\right)^T \left(H_{F \to s}(f_o) - H_{\mathbb{R} \to s}(x_b)\right)}_{Image\ cost\ J_I} , \tag{5}$$

     where $J_B$ and $J_O$ denote the background and observation terms in the conventional cost function, and $J_I$ represents the added

image observation term. In the image observation term $f(t) \in F$, $f_o$ represents the frame-wise observation image of the image dynamic observation system at moment $t_0$, which belongs to the image observation space $F$. $s$ represents the image space under the mathematical definition. The image structure operator $H_{F \to s}$ represents the mapping from the image space to the structure-defined mathematical space, that is, the structural information extracted from the image observations, which represents the multi-scale geometric features of the image. The operator $H_{\mathbb{R} \to s}$ implies a mapping from the space of state

variables to the mathematical space where the structure resides, which represents obtaining the same type of structural information from the background field output by the model.

## 2.3 Curvelet multiscale analysis method

From the above equations, it is clear that the image structure operator is a key technique for the image assimilation system. The important information in an image is mainly located in the discontinuities of the image, which can be well described by the spectral space, and thus the image can be quantitatively described by the spectral transformation coefficients of the image. The curvelet transform is exactly a suitable multi-scale transform analysis method, which is not only capable of time-frequency analysis, but also has strong directional selection and discrimination capabilities (Titaud et al., 2010).

The curvelet transform takes the inner product of the basis function and the signal to achieve a sparse representation of the signal. In the two-dimensional space $R^2$, the spatial variable is denoted by $x$ and the parent wave is denoted by $\varphi_j(x)$. Thus, the curve wave coefficient is the inner product of the function $f \in L^2(R^2)$ and the curve wave $\varphi_{j,l,k}$:

$$c(j, l, k) := \langle f, \varphi_{j,l,k} \rangle = \int_{R^2} f(x)\overline{\varphi_{j,l,k}(x)}d_x, \tag{6}$$

where $j$, $l$ and $k$ are the scale, direction and position parameters, respectively, and the biggest difference with other spectral analysis methods is that the curvelet transform takes into account the direction parameter. The curvelet coefficients are anisotropic, which can efficiently represent the image edges and fully exploit the image features. The reconstruction equation of the coefficients is:

$$f = \sum_{j,l,k} \langle f, \varphi_{j,l,k} \rangle \varphi_{j,l,k} = \sum_{j,l,k} c(j, l, k)\varphi_{j,l,k} . \tag{7}$$

An important fact in the practical application of data assimilation methods is the presence of observation errors, which in this case is represented by the noise in the observed images. In order to effectively remove the image noise and extract the main structural features from the image, we can choose the curvelet coefficients at different scales. A simple "hard threshold" approach can achieve this goal, by setting the curvelet coefficients below the threshold (represented by $\sigma$) to zero. Denoising and key-feature selection can be achieved by adjusting the threshold value.

Figure 1 gives the structural information of the soil moisture image extracted by the curvelet analysis method under different threshold conditions on May 1, 2016 in East Asia. Figure 1a shows the spatial distribution of soil moisture simulated by the CoLM land surface model, it can be seen that soil moisture is low in northwest China and high in the south and east. When the threshold is 0.1 (Fig. 1b), the reconstructed image reproduces the low-value areas of soil moisture in Northwest China and the high-value areas in eastern and southern China, but only represents the large-scale spatial structure features of the raw image. When the threshold value is increased to 0.5 (Fig. 1c), the reconstructed image is definitely close to the original image, and the critical features of the reconstructed image, such as the dry zone in Xinjiang-Inner Mongolia and the wet area in southeast China and Siberia, are basically consistent with those of the original image. Only some small-

scale noise information, such as two dry zones located in northern Tibet and a wet zone near 59°N in northeastern Lake

Baikal, has been filtered out. It is shown that the multi-scale structural information of the image could be efficiently

extracted by the curvelet analysis method, which provides a basis for introducing the spatial structure information of the

observed data into the assimilation.

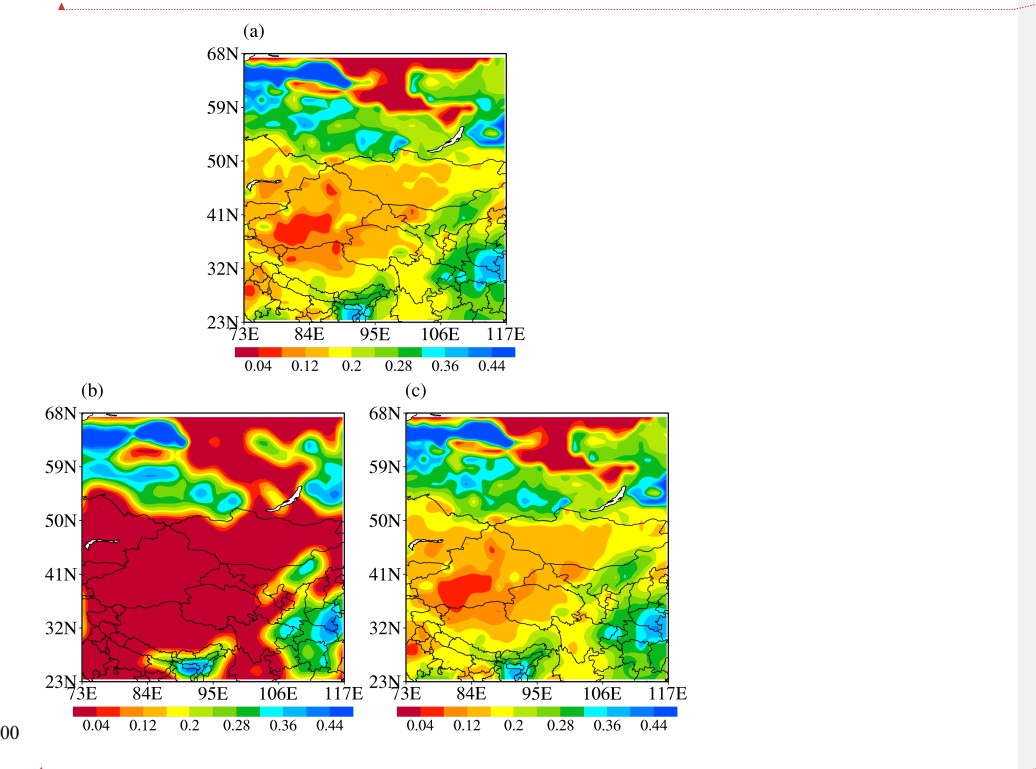

**Figure 1: Soil moisture distributions from (a) the raw image and extracted by curvelet analysis under the thresholds of (b) 0.1 and (c) 0.5 on May 1, 2016 in western East Asia.**

## 3 Experimental design and error analysis

### 3.1 Datasets

Taking the curvelet transform method as the image observation operator, we construct the land surface image assimilation system based on Eq. (5). To demonstrate the assimilation effect of the image assimilation system, the idealized data is used to examine the ability of the image assimilation method in adjusting the spatial structure of the land surface model variables. The soil moisture reanalysis data of the fifth generation ECMWF ReAnalysis Land (ERA5-Land) is chosen as the ideal observation data in this study, which has a horizontal resolution of 9 km and a temporal resolution of one hour. The H-TESSEL (Hydrology Tiled ECMWF Scheme for Surface Exchanges over Land) land surface model is used in ERA5-Land to output the soil volumetric water content data at four depth layers (0–7, 7–28, 28–100 and 100–289 cm). ERA5-Land is more accurate for all land use types with a series of improvements and recalculations based on ERA5. The calculations are made without coupling the atmospheric and wave module of ECMWF/IFS, allowing for a faster update frequency. The horizontal and temporal resolutions are respectively increased to 9 km and one hour, and the stratification of output soil volumetric water content is consistent with that of ERA5 (Sabater et al., 2021).

The soil volume water content reanalysis product V2.0, generated by the land surface data assimilation system CLDAS (CMA Land Data Assimilation System) of the National Meteorological Information Center of China Meteorological Administration, covers the Asian region (0–65°N, 60–160°E). The temporal resolution is 1 hour, and the spatial resolution is 0.0625°. The vertical direction is divided into five layers: 0–5 cm, 5–10 cm, 10–40 cm, 40–100 cm, and 100–200 cm. The CLDAS product is produced by using the near-real-time CLDAS atmospheric drive product, which incorporates a larger amount of ground station observation data and higher quality background field to drive various land surface models (such as CLM 3.5, CoLM and Noah-MP). As a result, the dataset exhibits excellent quality and offers high spatio-temporal resolution data in the China region (Shi et al., 2011; Liu et al., 2019). The CLDAS reanalysis data is therefore chosen as the independent dataset, and an additional verification analysis of the assimilation results based on the CLDAS data is conducted

### 3.2 Ideal experimental design

As shown in Fig. 1a, the selected region for the experiment (73°W-117°W, 23°N-68°N) covers most of the land area of China, and the model spatial resolution is 1.4° × 1.4°. The land-atmosphere coupling is the strongest in the western Qinghai-Tibet Plateau, where soil moisture has a large impact on the climate change and is an essential precursor signal for the summer precipitation forecasts in eastern China (Yuan et al., 2021). The western arid zone has complex topography, with strong spatial heterogeneity in soil moisture. In this region, the surface energy and water vapor budgets also have a crucial impact on the climate (Yang et al., 2021).

The assimilation is run from May 1 to August 31, 2016, and the prediction is made from September 1 to September 30, 2016. Two sets of experiments are designed. The first sets of experiments perform data assimilation (DA) four times a day with an interval of 6 hours (at 0000 UTC, 0600 UTC, 1200 UTC and 1800 UTC), and the soil moisture in the surface layer of 0–7 cm from ERA5-Land is assimilated. The other group is the control (CTRL) experiment, which has no observations assimilated.

Since it takes a period of time for the model to integrate to adapt to the soil moisture after assimilation, the first 15-day results of the experiment are discarded to ensure that the model can reach a new hydrological equilibrium state, which can make the evaluation of assimilation effect more objective. The analysis in this study mainly focuses on the period from May 16 to September 30, 2016. To highlight the effect of image assimilation, the other observations are not assimilated in this study, that is, $J_O$ is zero.

**3.3 Analysis of error characteristics**

From the cost function shown in Eq. (4), it can be seen that the solution of the cost function also requires the estimation of the background field error covariance and the observation error in advance, and the elimination of observation noise in the image. To obtain more precise analysis results, we need to accurately estimate the characteristics of various error covariances.

**3.3.1 The covariance matrix of background field error**

According to the characteristic of single-column model that there is no correlation between the simulation errors at different grid points, the covariance matrix of background field error can be directly expressed as the variance of the simulation error of the land-surface model at each grid point.

The node depths of the top three soil layers in the CoLM model are 0.70 cm, 2.79 cm and 6.22 cm, which are close to the first-layer depth of the ideal observation data (0-7cm). In this study, the hourly soil moisture data at the top three layers output by the land surface model from 2014 to 2015 is used as the background field. The soil moisture reanalysis data of the first ERA5-Land layer (0-7 cm) at the same time is interpolated to the depth corresponding to the background field, and then is used as the ideal observation data. Based on the difference between the background field and observations, the covariance matrices of background error are separately obtained for the soil moisture at each of the top three layers. The error covariance between different levels is not considered here.

Figure 2 shows the distributions of background error covariance for the top three soil moisture layers. It can be seen that the soil moisture in the top layer (Fig. 2a) is affected by myriad factors, so its background error is larger than that at the other two layers. The spatial characteristics of the background errors at different depths are similar, with relatively small errors in Xinjiang, northern Tibet and Inner Mongolia, which may be related to the drought in this region. In contrast, in humid areas such as Siberia, the soil moisture is affected by additional factors, which causes relatively larger simulation errors.

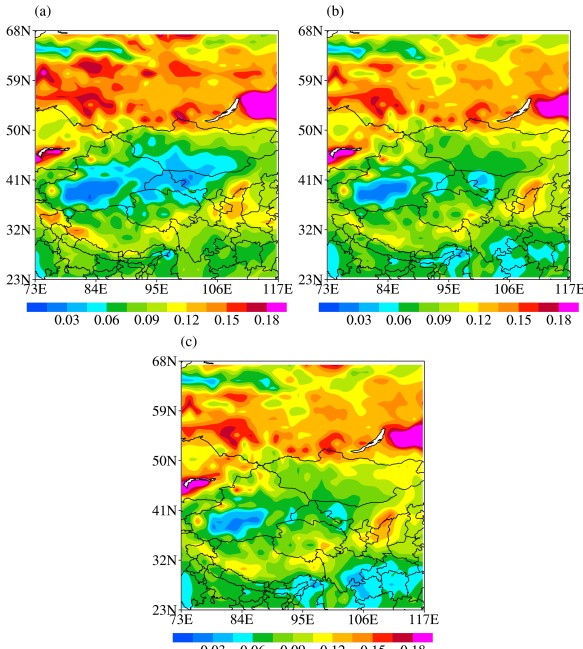

**Figure 2: The spatial distributions of soil moisture simulation error variance in the (a) first, (b) second and (c) third layers of the common land model (CoLM) during the statistical period from May 16 to September 30, 2016.**

### 3.3.2 Analysis of observation data error

In image assimilation, the observation noise can be efficiently eliminated by selecting an appropriate threshold value. To further objectively determine the threshold, the soil moisture of the ERA5-Land data at 100 instants is selected as the original image. Different threshold values are chosen for de-noising, and the reconstructed images with different degrees of de-noising are then obtained by inverse curvelet transformation. The threshold selection method is additionally discussed based on the statistical characteristics of the difference between reconstructed and original images.

Figure 3 shows the spatial distributions of the mean value of 100 reconstructed fields and the mean value of reconstructed errors, based on the raw soil moisture images every 6 hours (0000 UTC, 0600 UTC, 1200 UTC and 1800 UTC) from May 1 to May 25, 2016, with thresholds being 0.1 and 0.5 separately. As can be seen from the original image (Fig. 3a), in terms of large-scale structural features, soil moisture is relatively low in the central part of the selected region (from Xinjiang and northern Tibet to Mongolia), while it is relatively high in the surrounding of low-value areas. High-value centers of soil

moisture are found in the southern Siberian Plain, east of Lake Baikal, eastern China, and south of the Tibetan Plateau. When the threshold is 0.1, the average distribution of the reconstructed field (Fig. 3b) reproduces the large-scale characteristics of the original field that the low soil moisture is located in the middle and surrounded by high value centers. However, there are large errors between the reconstructed field and the original field (Fig. 3c). In particular, the spatial distribution of errors is similar to that of the large-scale original field, which indicates the loss of spatial structure information of the observations. When the threshold increases to 0.5, the spatial correlation coefficient (SCC) between reconstructed field (Fig. 3d) and original field is greater than 0.99, and the multi-scale features of the original field are properly reflected. As can be seen from Fig. 3e, the errors between the reconstructed and original fields are basically within 0.02, and the error distribution presents no obvious spatial structure characteristics.

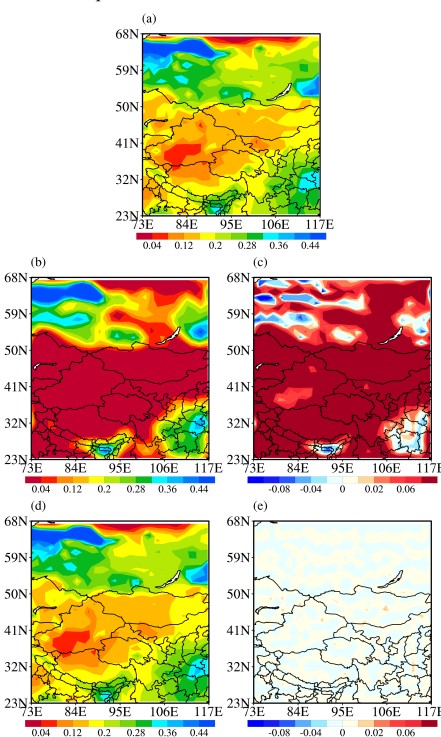

**Figure 3: The spatial distributions of (a) original soil moisture, (b, d) reconstructed soil moisture and (c, e) reconstruction errors under the threshold values of 0.1 (b, c) and 0.5 (d, e) averaged from May 1 to May 25, 2016.**

The image assimilation system finds the spatial structural characteristics of assimilation according to the threshold
values, and different thresholds could result in certain variations in assimilated spatial structure. In order to clarify the spatial
structure differences corresponding to different thresholds, the spatial correlation method (Daley, 1991) is employed in this
study to elucidate the distinctive characteristics of spatial structure corresponding to varying thresholds.

The hourly soil moisture data from ERA5-Land from May 1 to 30, 2016 is selected for analysis. The threshold σ means
the modulus of the decomposition coefficient falls within the first $100*\sigma\%$ percentile. For instance, a value of 0.5 indicates
that the mode retaining the top 50% of decomposition coefficient. The original image can be reconstructed by selecting
different threshold ranges, namely (0,0.01], (0.01,0.03], (0.03,0.05], (0.05,0.1], (0.1,0.2], (0.2,0.3], (0.3,0.4], (0.4,0.5],
(0.5,0.6], (0.6,0.7], (0.7,0.8], (0.8,0.9] and (0.9,1.0]. The correlation coefficient between each grid point and its neighboring
grid points can be obtained based on the reconstructed time series of each grid point. The spatial structural characteristics of
different scales in the reconstructed images could be quantitatively expressed by the average correlation coefficients
corresponding to different grid point distances.

The mean correlation coefficient corresponding to grid point distance is illustrated in Figure 4. As can be seen, the
variation characteristics of the inter-grid correlation coefficient of the original soil moisture represented by the black line
with respect to the grid distance. The average correlation coefficient can exceed 0.5 within a radius of 200 km, while
maintaining above 0.4 within a radius of 300 km. The distance corresponding to high correlation coefficients represents the
characteristics of consistent changes in soil moisture within a similar range, that is, soil moisture has the characteristics of
spatial structure at the corresponding scale.

When the threshold value is 0.01, the average correlation curve exhibits a similar change in correlation coefficient of
the original variable, thereby indicating that the curvelet coefficient corresponding to this threshold value effectively
reproduces the large-scale spatial structure. The spatial structure scale represented by the corresponding curvelet
transformation reconstruction results decrease as the threshold value increases, leading to a rapid decrease in the correlation
coefficient with increasing distance. The curvelet reconstruction results with different threshold intervals represent the
structural characteristics of different horizontal scales, while the cumulative threshold can well represent the spatial
structural characteristics of soil moisture variables represented by the selected threshold in the assimilation. The average
correlation coefficient of the cumulative threshold is depicted in Figure 4b. As can be seen, the top 10% of curvelet
coefficients can effectively replicate the spatial correlation characteristics of soil moisture variables. The results also indicate
that the variations in threshold values have minimal impact on the assimilated spatial structure when the threshold value
exceeds 0.1.

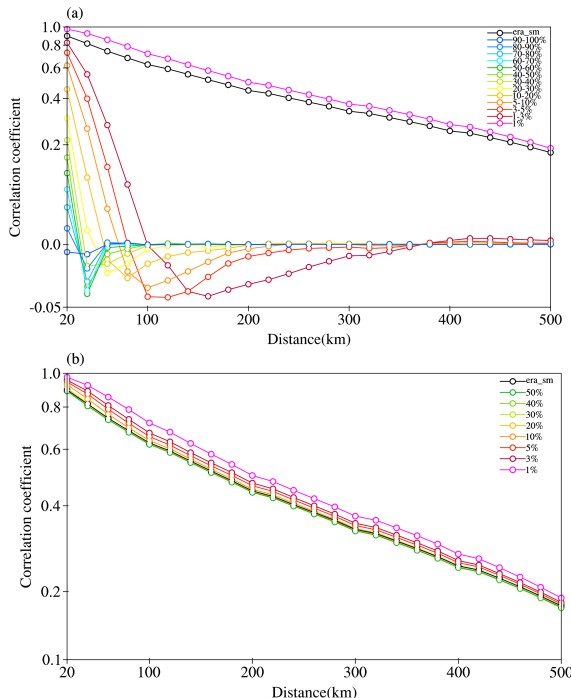

**Figure 4: Variation curves of the average correlation coefficient between grid points with the distance in the reconstructed ERA5-Land hourly soil moisture image of the study area from May 1 to 30, 2016, which is reconstructed based on the curvelet coefficients of (a) different threshold intervals and (b) cumulative thresholds.**

Naturally, a higher threshold can effectively capture more spatial structural features of the observed variables, but the presence of observation errors imposes limitations on its continuous increase. The observational error is typically characterized by stochastic fluctuations. When the discrepancy between the reconstructed results and the original variables exhibits random variation characteristics, it can be inferred that the observation information eliminated by the threshold method primarily consists of observation errors.

To better clarify the statistical characteristics of the reconstruction errors under different thresholds, Figure 5 shows the probability density distribution curves of the reconstruction errors for 100 reconstructed fields at different thresholds. For the error at the threshold of 0.5, the skewness coefficient of the probability density distribution curve is 0.00 and the kurtosis coefficient is 0.38, indicating the curve is close to the standard normal distribution curve (the skewness and kurtosis

coefficients are all 0). With the gradual increase of threshold value, although the reconstruction error decreases, the residual error is mainly concentrated in the range of smaller values, and the curve shows a "sharp peak" distribution. Considering that the observation errors are mostly random errors, it is reasonable to believe that the reconstruction errors at the threshold of 0.5 are mainly observation errors, which also implies this threshold is good for the purpose of de-noising the observation images.

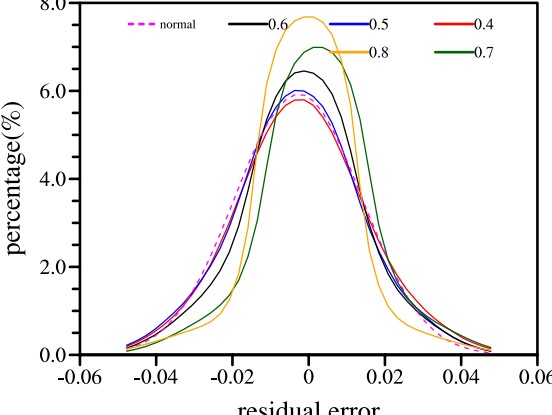

**Figure 5: Probability density distributions of 100 reconstructed errors under different thresholds. The magenta dashed line represents standard normal distribution. The red, blue, black, green and orange solid lines represent threshold values of 0.4, 0.5, 0.6, 0.7 and 0.8, respectively.**

### 3.4 The influence of image assimilation constraints

Two sets of ideal experiments are designed to validate the impact of image assimilation and evaluate its superiority over traditional single column assimilation in adjusting the spatial distribution structure of soil moisture. The ideal observational data for assimilation is the ERA5-Land reanalysis soil moisture. The first set corresponds to the conventional assimilation experiment, where $J_{(x)} = J_B + J_{Os}$ as described by Equation (5). Another set is image assimilation experiment, where $J_O = 0$ in Equation (5), indicating that $J_{(x)} = J_B + J_{Ii}$.

The process of data assimilation entails leveraging the discrepancy between observed data and background field, in conjunction with a priori knowledge of observation error and background error, to derive an analysis field that closely approximates the true value. The primary challenge in single column assimilation lies in acquiring precise prior information

regarding observation error. The spatial distribution of observation error for a specific single column assimilation experiment is illustrated in Fig. 6. In consideration of the necessity for an ideal experiment, it is assumed that the observation error outside the China region is negligible, while a significant error is presumed within the China region, so as to emphasize the impact of observation error on assimilation results.

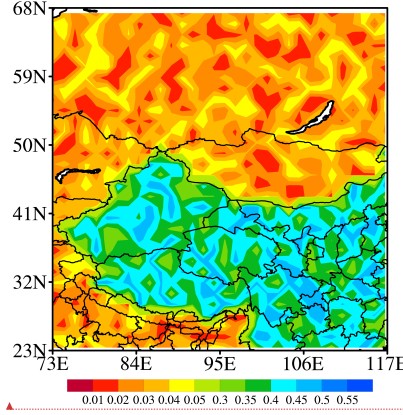

**Figure 6: Spatial distribution of observation errors.**

The spatial distributions of soil moisture for the ideal observation data and different experiments at 0000 UTC on May 1, 2016 are given in Figure 7. The spatial distribution of surface soil moisture in ERA5-Land is illustrated in Fig. 7a. The northern Siberian region of the selected area exhibits a relatively high soil moisture content overall, with a ring-shaped distinct wet zone in the northwest. The central region stretching from Xinjiang to western Mongolia is a significant arid area. However, the soil moisture in the Tianshan Mountains is wet. The soil moisture of the Qinghai-Tibet Plateau region gradually decreases from west to east. The soil moisture in southern Qinghai, Hunan and Jiangxi is characterized by high level of saturation, while Gansu, Ningxia and Hebei experience relatively arid soil conditions. Figure 7b is the distribution of soil moisture in the control experiment (background field). It is evident that there are significant disparities in the spatial distribution of soil moisture when compared with the reanalysis data. In the control experiment, a dry region extends from west to east in the northern area of Lake Baikal, while eastern Kazakhstan and central Inner Mongolia also exhibit arid conditions.

Figure 7c shows the results of the single point assimilation experiment. The observation error outside the China region is relatively minimal, indicating a strong correspondence between the analysis field and the observation data, and the overall distribution also exhibits a high degree of conformity with the observation. The analysis field in China region, however,

closely resembles the background field. Nevertheless, there is a significant disparity between the observed soil moisture and that of the background field, indicating a lack of adjustment based on observed information.

Figure 7d is the assimilation results of the image assimilation ideal experiment. It is evident that image assimilation effectively adjusts the distribution pattern of soil moisture. The above-mentioned characteristics of moist soil moisture in the northwest region of the observation field, the arid region of Xinjiang and Mongolia, and the humid region of the Tianshan Mountains are all well reflected in the analysis field.

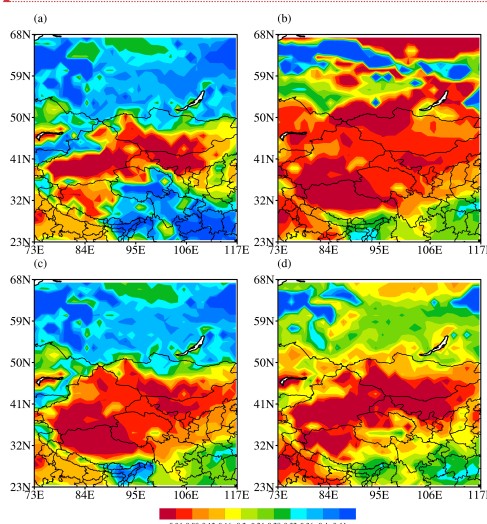

**Figure 7: Spatial distributions of surface soil moisture for (a) ERA5-Land, (b) CTL experiment, (c) analysis field of conventional assimilation experiment, and (d) analysis field of image assimilation experiment at 00:00 UTC on May 1, 2016.**

## 4 Results

Figure 8 shows the variation of the cost function values with the number of iterations when assimilating ERA5-Land surface soil moisture using the image assimilation system at 0000 UTC on May 16, 2016. The criterion of convergence is that the gradient of the cost function values is less than $10^{-9}$. It can be seen that the initial value of the cost function is 1121.0, which has been reduced to 863.1 by the second iteration. The convergence speed of the cost function is relatively fast, and it only

needs 22 iterations, which also demonstrates the validity and rationality of introducing the image operator term in the cost function. The fast convergence speed of the cost function caused by the constraint of the image operator also indicates that the assimilation can effectively absorb the spatial structure information of the observation.

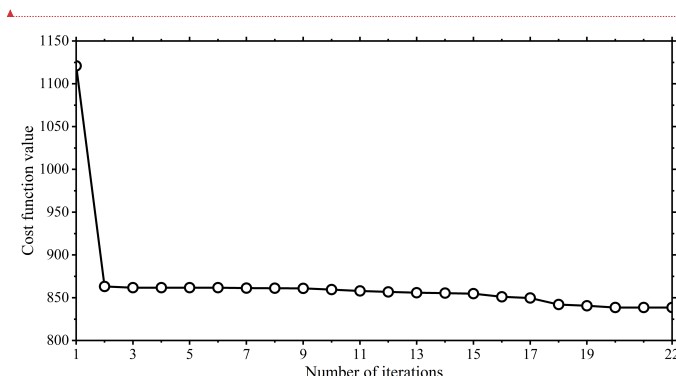

Figure 8: Variation of the cost function value with the number of iterations using the image assimilation system at 0000 UTC on May 16, 2016.

Figure 9 shows the 0-7 cm soil moisture distributions of the observation, the analysis field from the image assimilation system and the output from CoLM at 0000 UTC on May 16, 2016. It can be seen that the surface soil moisture of ideal observations (Fig. 9a) is drier along Mongolia and Xinjiang, but wetter in southern China. The observed soil moisture is also relatively high in the vicinity of the Tianshan Mountains, the eastern part of Qinghai-Tibet Plateau and the southern part of Lake Baikal, as well as in the eastern parts of Henan Province and Inner Mongolia. However, the results of the CTRL experiment (Fig. 9c) show a "high-low-high" distribution from south to north, which is extremely different from the spatial structure of the observation. It can be seen from the spatial distribution of the analysis field that the soil moisture structures are all remarkably improved at different scales (Fig. 9b). In view of large-scale structure, the structural wet bias from northwest China to Mongolia can be thoroughly corrected, and the observed structural features of low soil moisture from southern Qinghai to southeastern China are nicely reproduced in the analysis field. In addition, some relatively small-scale structures, such as the relatively wet soil in the Tianshan Mountains region of Xinjiang, the central part of Qinghai Province, and the northeastern part of Lake Baikal, are also well represented in the analysis field.

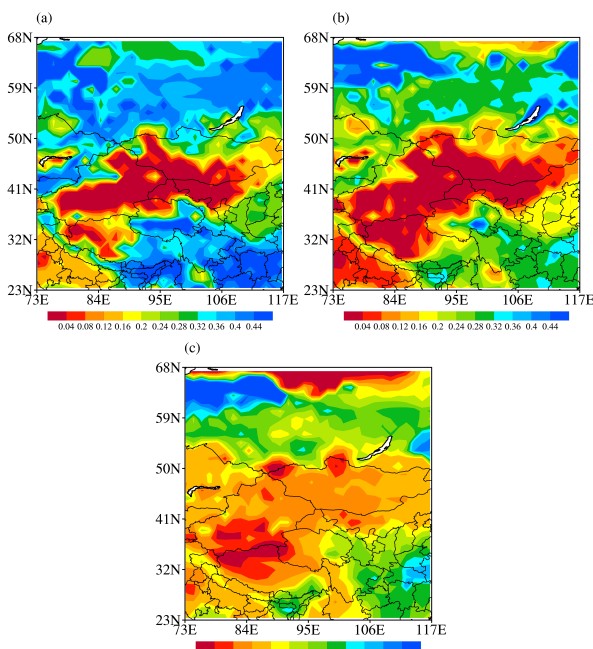

**Figure 9: Spatial distributions of soil moisture at the depth of 0-7 cm for (a) the observation, (b) the analysis field from the image**
**assimilation system and (c) the background field from the CoLM at 0000 UTC on May 16, 2016.**

Figure 10 shows the spatial distributions of soil moisture at 0-7 cm at 0000 UTC on September 1, 2016 from the observation, DA experiment and CTRL experiment. It can be seen from Fig. 10 that, after four consecutive months of cyclic assimilation, the large-scale struct ure of surface soil moisture in the analysis field is much closer to the observation than that
in the CTRL experiment. The improvement is mainly concentrated in the dry zones along Xinjiang-Mongolia, as well as the wet centers in the Tianshan Mountains and central-northern Qinghai Province. The dry tongue from Ningxia Province to Shanxi Province in China has also been reproduced in the analysis field. In addition, the distribution structure of wet areas in the region north of 60°N and east of 111°E has also been thoroughly improved in the analysis field. Overall, the structural characteristics of soil moisture at different scales in the analysis field are in better agreement with the observations.

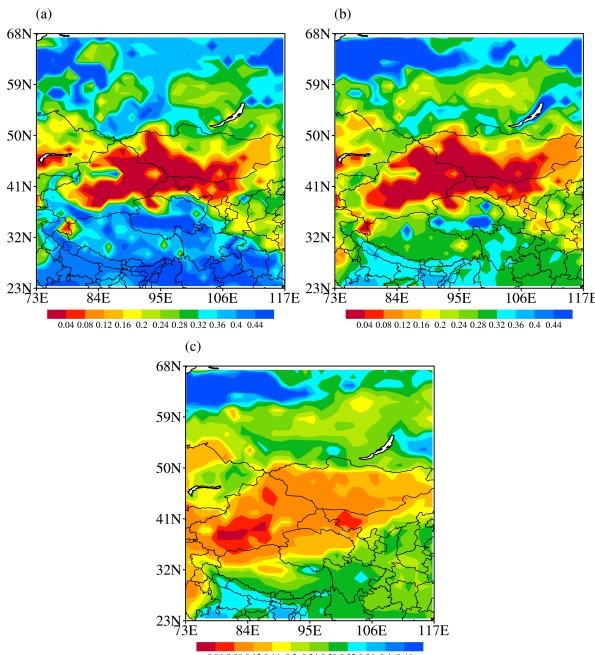

**Figure 10: The soil moisture at 0-7 cm from (a) the observation, (b) the data assimilation (DA) experiment after 4 months of continuous assimilation and (c) the control (CTRL) experiment at 0000 UTC on September1, 2016.**

Moisture condition of the underlying surface may have some influence on short-term climate anomalies, but whether the effect is significant mainly depends on the duration of the underlying surface features. The surface soil moisture is considerably affected by external high-frequency perturbations, so the retained anomalous signals are susceptible to interference, making the anomalous signals difficult to maintain. However, the deep-layer soil moisture has an excellent and persistent ability to maintain the abnormally strong signals,, which may have a certain impact on the later climate anomalies (Xu et al., 2021). Therefore, it is necessary to conduct further analysis on soil moisture improvement at a deeper level through image assimilation system.

By assimilating the surface soil moisture through the image assimilation system, the deep-layer soil moisture is simultaneously adjusted under the soil hydrodynamic and thermodynamic constraints of the land surface process model. Figure 8 shows the spatial distributions of the ideal soil moisture observation and the soil moisture predictions from the DA and the CTRL experiments at the depth of 7-28 cm at 0000 UTC on September 1, 2016 after the final assimilation.

**Deleted:** b

It can be seen that the distribution pattern of deep-layer soil moisture observation (Fig. 11a) is relatively consistent with that of the surface-layer soil moisture. This essentially shows that the region from Xinjiang to Mongolia is an arid region, while the relatively high soil moisture regions are located in southern-southeastern China, and the Siberian Plain. The high-value centers for the surface and soil moisture content at a depth of 7–28 cm are practically the same, but overall, the soil moisture at a depth of 7–28 cm is wetter than the surface. The analysis field of image assimilation (Fig. 11b) shows drought in the southern part of Mongolia, which is consistent with the observation. At the same time, the high-value centers near Novosibirsk, the Tianshan Mountains and the central part of Qinghai Province are also well reproduced. It indicates that the image assimilation also has excellent results in improving the spatial structure of soil moisture at deeper layers.

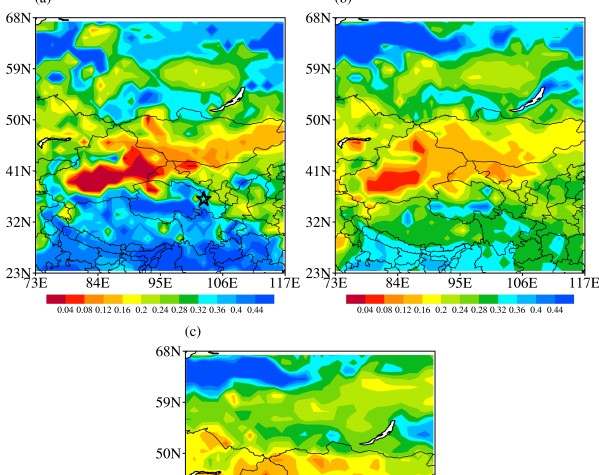

**Figure 11: Same as Fig. 10, but for the soil moisture at the depth of 7-28 cm. The pentagram shows the station location for the single-site analysis in Fig. 15.**

Vertical motion of soil water is integrated over the layer thickness, in which the time rate of variation in water mass must equal to the net flow across the bounding interface, and plus the rate of internal source or sink. The terms of water flow across the layer interfaces are linearly expanded by using first-order Taylor expansion. Therefore, when the surface data were assimilated, the net flow across the bounding interface to deeper layers become more reasonable corresponding to

surface variation. Of course, when it comes to the process of permafrost and snow processes, such as soil freezing and thawing in the Tibetan Plateau region, the variations of soil moisture are much more complex, and the mechanism of data assimilation on permafrost needs to be studied more thoroughly in the future.

In order to further elucidate the vertical impact of data assimilation, the vertical propagation characteristics of surface assimilation influence are also examined based on actual experiment results. The vertical-temporal profiles of soil moisture on different underlying surface types selected in the Tibetan Plateau and plain areas are given in Fig. 12, so as to elucidate the physical processes how the surface soil moisture assimilation influences soil moisture at a depth of 7–28 cm. The spatial locations of selected single points are depicted in Fig. 12a. In order to emphasize the soil moisture variation difference

between plateau areas and plain areas, bare soil points are situated in the eastern and western regions of the plateau (represented by blue and black five-pointed stars), while corn and needleleaf evergreen boreal tree (represented by red and orange five-pointed stars) are positioned within the plain area. Figs. 12b–12c illustrate the difference of soil-moisture analysis field between DA experiment and CTL experiment, as well as the temporal characteristics of soil moisture analysis field at different depths of selected points in plateau areas. The vertical ordinate denotes the position of node depth for each

soil layer in the CoLM model. The most notable difference in the vertical variation of soil moisture among the two points on the plateau is primarily attributed to the differences in both magnitude and depth of this vertical change. In the western plateau region, soil moisture at bare soil points is generally low, usually below 0.2 $m^3/m^3$ (Fig. 12b). Additionally, the surface undergoes significant temporal variations that may be related to the prevalence of small-scale convective weather systems in this plateau area. The vertical variation of bare soil moisture in the plateau region primarily occurs above 50 cm,

while the soil moisture exhibits a consistent pattern below 50 cm. The vertical variation of soil moisture is correlated with the intensity of soil moisture anomaly. As depicted in Figs. 12b and 12c, the vertical impact of minor perturbations in bare soil moisture within the plateau region is negligible, primarily occurring above a depth of 3 cm. The similarity between the two bare soil points lies in the fact that significant changes in soil moisture can rapidly impact the top 10 cm of soil, resulting in similar characteristics observed in the soil moisture above this depth. However, abnormal soil moisture exhibits a

noticeable time lag effect below 10 cm. The characteristics of assimilation influence exhibit similarities to the features observed in vertical changes of soil moisture. Assimilation significantly enhances surface soil moisture around July 10th, and the increasement in soil moisture analysis within the plateau region can also rapidly impact the 10 cm depth of soil, with a maximum positive analysis increment reaching 0.16 $m^3/m^3$. The impact of assimilation can affect soil moisture at a depth of approximately 10 cm within one day, while it takes approximately 15 days for this analysis to affect the 50 cm depth.

However, the impact of the analysis increment can be sustained for over a month at the depths ranging from 20 cm to 50 cm.

       Figures 12d and 12e are similar to Figs. 12b and 12c, but they are selected from the plain areas. It is evident that the vertical variation characteristics of soil moisture differ significantly among different vegetation types. The analysis increment for corn is relatively minimal. Image assimilation leads to a substantial increase in surface soil moisture around July 10th. The maximum positive analysis increment can reach up to 0.12 $m^3/m^3$, with a vertical change level reaching

approximately 30 cm. The effect is gradually transmitted to a depth of approximately 2 meters over time, with a duration of about one month. In the case of needleleaf evergreen boreal tree, the analysis increment is relatively small, and surface soil moisture gradually increases from around July, with its influence extending to a depth of approximately 100 cm. Seen from the above analysis, it is evident that the assimilation of surface soil moisture gradually impacts the deeper layers of the model as integration progresses, with a lasting effect of approximately 1 month. This phenomenon also serves as the primary

factor contributing to the simulation improvement of soil moisture at a depth of 7–28 cm.

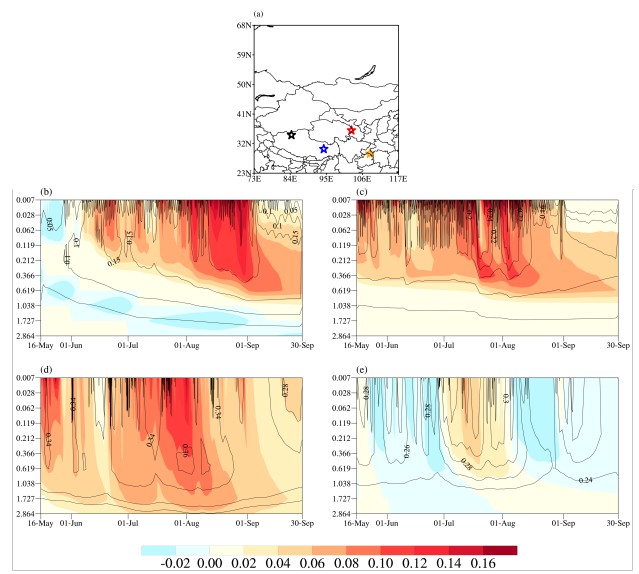

**Figure 12: (a) The location of designated grid. The soil-moisture temporal variation of the difference between the DA experiment and CTL experiment (represented by shadow) and the soil moisture profiles (indicated by contours) under different land types: (b) bare soil (black five-pointed star), (c) bare soil (blue five-pointed star), (d) corn (red five-pointed star), and (e) needleleaf evergreen boreal tree (orange dots).**


To quantitatively assess the effect of image assimilation in improving the spatial structure of model soil moisture, the SCCs between the ideal observation and model outputs with and without assimilation are calculated and shown in Fig. 13. It can be seen that the SCC of the DA experiment is already much higher than that of the CTRL experiment after the first assimilation, with the SCC increased from 0.44 to 0.76. This proves that the image assimilation can quickly and effectively

adjust the spatial structure of soil moisture. During the period of cyclic assimilation period, the SCC can be maintained above 0.6 with an average value of 0.67, which is steadily higher than that of the CTRL experiment. This indicates the image

assimilation effectively improves the spatial structure of soil moisture, making it more consistent with the observation. The SCC of the DA experiment is also higher than that of the CTRL experiment in the one-month prediction stage after September 1, indicating that the optimization of the soil moisture spatial structure by image assimilation could have a obvious positive impact on the prediction in the following month.

Soil moisture is relatively more stable at subsurface depths. The SCC of subsurface soil moisture between observation and CTRL experiment is 0.31 at the initial time, which increases to 0.53 by introducing image assimilation. After cyclic assimilation, the mean value of the SCC between observation and DA experiment increases from 0.41 to 0.57, which is higher than that between observation and the CTRL experiment throughout the entire assimilation period. This indicates that by optimizing the spatial structure of the soil moisture in the surface layer, the soil moisture in the deeper layers is also favorably improved. In the prediction stage, the SCCs between the DA experiment and the observation are always higher than those between observation and the CTRL experiment. The mean value reaches 0.63, suggesting that optimization of surface soil moisture could lead to an excellent improvement in the prediction of deep-layer soil moisture.

It is important to note that the SCC exhibits a clear temporal variation, which does not necessarily imply a time-varying assimilation effect. This can be attributed to the dominant influence of precipitation on the changes in the SCC. Hence, Figure 13 also includes the hourly total precipitation (represented by grey bars) in the model domain. The changes in precipitation exhibit a strong correlation with the SCC. From May 16 to June 15, there is minimal precipitation, corresponding to sustained high SCCs of soil moisture (red line) after assimilation. Subsequently, as precipitation increases, the SCC gradually diminishes. From August 15 to September 1, the SCC exhibits an inverse variation with decreasing precipitation.

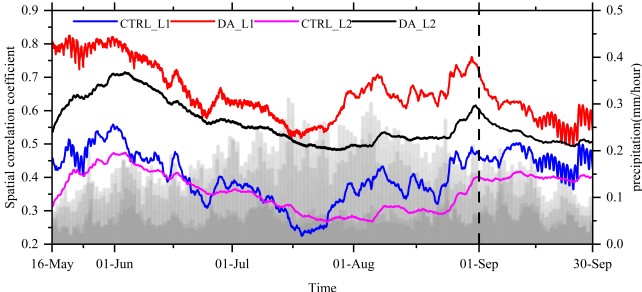

**Figure 13: Hourly variations of the spatial correlation coefficient of the surface (red and blue solid lines) and subsurface (black and magenta solid lines) soil moisture between the observations and the experiments with (red and black solid lines) and without (blue and magenta solid lines) image assimilation, and the precipitation in the forced data was indicated by grey shading. After the vertical dashed line, it is the prediction period.**

The overlapping region (22–50°N,73–117°E) between the CLDAS data and the model region is selected for analysis.

The spatial correlation coefficients of soil moisture before and after assimilation to CLDAS data are also computed, aiming to quantitatively assess the accuracy of the adjustment in soil moisture spatial distribution structure by the image assimilation system. The image assimilation results in a notable increase in the spatial correlation coefficient between CoLM soil moisture and the first layer (0–5cm) soil moisture of CLDAS, as depicted in Fig. 14a. Throughout the assimilation and prediction stages, this correlation coefficient consistently surpasses that of the CTL experiment, with a maximum value of

0.79. Moreover, after assimilation, there is an average increase in spatial correlation coefficient from 0.67 to 0.71. The image assimilation brings a more significant increase in the spatial correlation coefficient of soil moisture in the second layer (5–10 cm), as depicted in Fig. 14b. The highest spatial correlation coefficient reaches 0.79, while the average value increases from 0.67 to 0.73. The verification results of independent data further confirm that the image assimilation system has a strong capability in adjusting the spatial structure of soil moisture, particularly in relation to subsurface soil moisture.

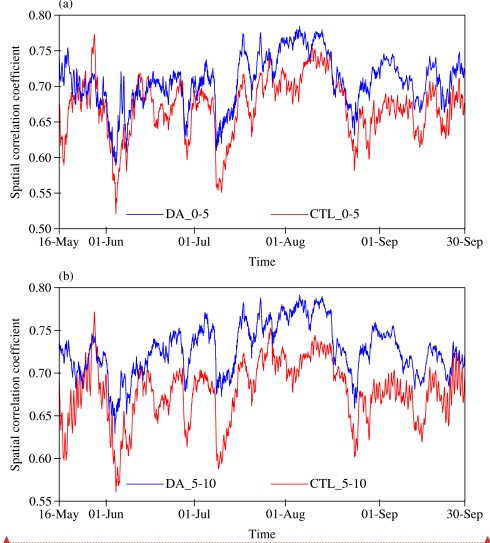

**Figure 14: The spatial correlation coefficients of the CLDAS products to the CTL experiment (red solid line) and image assimilation experiment (blue solid line) for the first layer (0–5cm) and second layer (5–10 cm) from May 16 to September 30, 2016.**

In order to further show the variation characteristics of soil moisture during assimilation, a single-point analysis is also performed by using the hourly soil moisture data from observation, DA and CTRL experiments at a single station in the Tianshan Mountains region of Xinjiang Province. From the hourly variation of 0-7 cm soil moisture (Fig. 15a), it can be seen that the observed soil moisture fluctuates around 0.40 while the soil moisture of the CTRL experiment has an obvious deviation from the observation, and they have different variation trends. However, the soil moisture slowly adjusts during the image assimilation period and gradually approaches the observation from mid-May to mid-June. By late June, the surface soil moisture gradually increases to more than 0.33 $m^3 \cdot m^{-3}$, which is closer to the observation. At the prediction period in September, the soil moisture in the DA experiment is also closer to the observation than that in the CTRL experiment.

From the hourly variations of soil moisture at 7-28 cm (Fig. 15b), it can be seen that the observed soil moisture in the deeperlayer is more stable than that in the surface layer, and the variation range is smaller, but the trend in the deeper layer is approximately the same as that in the surface layer. In late May, when the surface soil moisture becomes wetter, the deep-layer soil moisture in the DA experiment gradually responds, and its value gradually increases and approaches the observed value. Both in the assimilation and the prediction periods, the soil moisture in the DA experiment is closer to the observation than in the CTRL experiment.

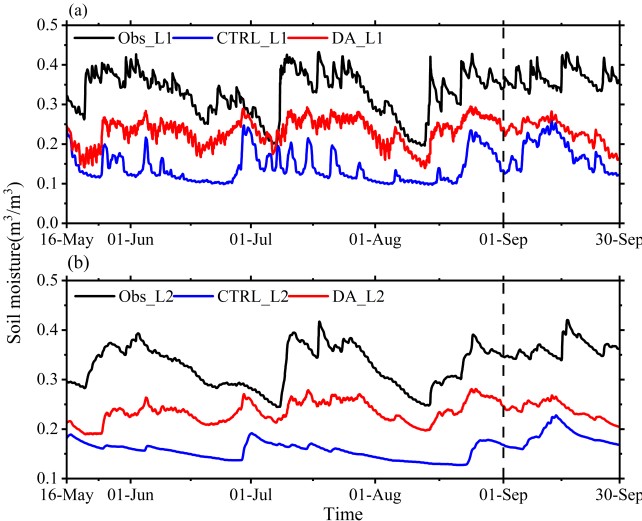

Figure 15: Hourly variations of the (a) 0-7 cm and (b) 7-28 cm soil moisture in the observation (black solid lines), DA experiment (red solid lines) and CTRL experiment (blue solid lines) at a single station in the Tianshan Mountains region of Xinjiang Province from May 16 to September 30, 2016. The prediction period is after the vertical dashed line.

The improvement of soil moisture after image assimilation is further evaluated based on the root mean square error (RMSE). Figure 16 shows the hourly RMSE variations of surface and subsurface soil moisture in DA and CTRL experiments. As can be seen, the RMSE of surface soil moisture in the CTRL experiment is larger with mean value of 0.16 $m^3 \cdot m^{-3}$, and the RMSE fluctuates considerably due to the influence of additional factors. The RMSE is fundamentally reduced by about 0.04 $m^3 \cdot m^{-3}$ after image assimilation, which also indicates that the image assimilation not only optimizes

the spatial distribution structure of soil moisture, but also has a certain improvement effect on the soil moisture values. In the prediction period, the surface layer is more disturbed by atmospheric forcing, so the RMSE at surface layer gradually increases with time, but the RMSE of DA experiment is also consistently smaller than that of the CTRL experiment.

The RMSEs of subsurface soil moisture in both experiments are smaller than that of the surface soil moisture. Although the initial error is larger, but it gradually decreases with time and shows a stable variation. The mean RMSE of the

subsurface soil moisture in the CTRL experiment is about 0.15 $m^3 \cdot m^{-3}$, while it reduces to 0.12 $m^3 \cdot m^{-3}$ after assimilation. Similarly, the RMSEs of the DA experiment are steadily less than those of the CTRL experiment in both cyclic assimilation and prediction periods.

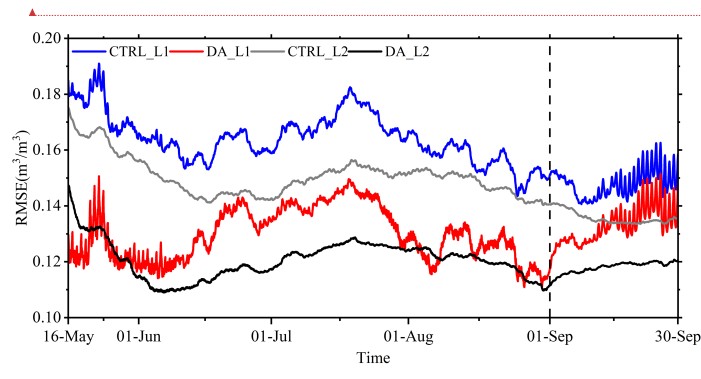

**Figure 16: Hourly variations of the soil moisture RMSEs for the DA (red and black solid lines) and CTRL experiments (gray and blue solid lines) at the surface (blue and red solid lines) and subsurface layers (black and gray solid lines) in the study area.**

**5 Discussion and conclusions**

The exchange of heat and water vapor between the land surface and the atmosphere plays a crucial role in influencing weather and climate change. The impact of soil moisture on atmospheric changes is frequently manifested through the persistent influence of large-scale soil moisture anomalies. The construction of an assimilation system with image

assimilation capability is aimed at enhancing the spatial structure accuracy of soil moisture anomalies in the initial field of land surface models. The system is primarily based on the three-dimensional variational data assimilation framework, employing the curvelet transformation method with multi-scale transformation capability and anisotropic basis function as the observation operator. By incorporating image structural similarity as a weak constraint in the cost function, the spatial structure of soil moisture in the initial condition is effectively adjusted to align with the structural characteristics of observed soil moisture image, thereby enhancing the accuracy of soil moisture simulation.

The performance of the image assimilation system is systematically validated by conducting ideal experiments, with the ERA5-Land reanalysis data as ideal observations, and the CLDAS reanalysis product is incorporated for independent verification. The findings demonstrate that the assimilation of surface soil moisture observation images effectively and reasonably enhances the spatial structure of soil moisture analysis field. The spatial correlation coefficient between the analysis and ERA-Land reanalysis data increases significantly from 0.39 to 0.67, while the root-mean-square error decreases notably from 0.16 m³/m³ to 0.12 m³/m³. With the improvement of surface soil moisture, the spatial pattern of subsurface soil moisture is further optimized under the reasonable constraints of model dynamics and thermal processes. There is an increase (from 0.35 to 0.57) in the spatial correlation coefficient between the soil moisture at a depth of 7–28 cm and the ERA-Land data. The root mean square error decreases from 0.15 m³/m³ to 0.13 m³/m³.

The verification results based on independent data CLDAS consistently demonstrate a higher spatial correlation coefficient between CoLM surface (0–5 cm) soil moisture in the assimilation experiment and the CTL experiment, with a maximum correlation coefficient of 0.79 throughout both assimilation and prediction stages. The average spatial correlation coefficient for surface soil moisture increases from 0.67 to 0.71 after image assimilation. While for subsurface (5–10 cm) soil moisture, it steadily rises from 0.67 to 0.73 on average. These quantitative evaluation outcomes fully validate the practical applicability of the new image assimilation method.

The image assimilation system developed by this study could effectively optimize the spatial structure of soil variables in the background by incorporating constraint conditions of the observed spatial structures. The method demonstrates excellent applicability to various soil variables, effectively mitigating the negative impact of strong spatial heterogeneity of soil on data assimilation. The key challenge in image assimilation lies in obtaining accurate spatial structure observation of soil variables. The data of ground automatic stations with high spatial-temporal resolution established in China, along with satellite observation data that can overcome natural constraints and achieve large-scale uniform observation in various terrains, are capable of providing observational images depicting the spatial structure of land surface variables for image assimilation. The effective assimilation of the spatial structural characteristics of those high-density meteorological observation data, is the primary focus of our subsequent research. However, how to establish the direct spatial structure relationship between satellite-observed brightness temperature data and soil variables, and how to repair these non-uniform data into uniformly distributed data, these are the key technical problems that need to be solved in the future.

Additionally, it should be noted that the image assimilation method and the prevailing single-point land assimilation method in current practice are not mutually exclusive. The single-point land assimilation method is more suitable for

assimilating sparse observation data in key areas. However, if the image assimilation method is used to optimize the fine structure of soil moisture in specific areas, the threshold σ mentioned above needs to be further increased, but this approach is susceptible to introducing additional observational errors. Therefore, by integrating the capacity of the image assimilation method in adjusting the large-scale spatial structure of soil variables and the capability of single-point land assimilation method in finely optimizing soil variables in crucial regions, and by leveraging the advantages offered by diverse types of meteorological observation data, we can attain more refined initial conditions for land models, which constitutes the primary objective of our subsequent research.

*Code and data availability.* The code of the Common Land model (CoLM) version 2014 was obtained from http://globalchange.bnu.edu.cn/research/models (Ji et al., 2014). The atmospheric forcings and CoLM rawdata for making land surface data are also available at http://globalchange.bnu.edu.cn/research/models (Qian et al., 2005). The ECMWF ERA5-Land hourly data from 1981 to present (Muñoz Sabater et al., 2019) were acquired from the Copernicus Climate Change Service (C3S) Climate Data Store (https://doi.org/10.24381/cds.e2161bac, last access: 11 February 2022). The code of the Common Land model (CoLM) version 2014, and the source code of the newly image data assimilation sysytem, as well as the data process software codes, and the model outputs data have been uploaded to Zenodo repositories, which are available at https://doi.org/10.5281/zenodo.10068298 (Shen, W., 2023).

*Author contributions*. Conceptualization, Z.Q. and W.S. and Z.L.; methodology, W.S.; software, W.S. and Z.Q.; validation, Z.Q., W.S. and Z.L.; formal analysis, J.L.; resources, Z.L. and J.L.; data curation, W.S.; writing—original draft preparation, W.S.; writing—review and editing, Z.Q. and J.L.; visualization, W.S.; supervision, Z.L. and J.L.; project administration, Z.Q.; funding acquisition, Z.Q. and Z.L. All authors have read and agreed to the published version of the manuscript.

*Acknowledgements.* This research was supported by the National Key Scientific and Technological Infrastructure project "Earth System Numerical Simulation Facility" (EarthLab).

*Financial support*. This research has been supported by the National Natural Science Foundation of China (grant no. 42075166 and 42375004); the Postgraduate Research and Practice Innovation Program of Jiangsu Province (KYCX23_1306); the Fengyun satellite ground engineering application project (FY-3(03)-AS-11.08).

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
