# Peer review of "Development and preliminary validation of a land surface image assimilation system based on the common land model"

_EGUsphere, 2023_

## Author Comment (AC2)

**Summary:**

Most current land surface assimilation systems are basically single point assimilation. Single point assimilation can easily break the coherent large-scale spatial structures of soil moisture anomaly, which are usually the important land surface factors to influence short-term climate variations over the land. The manuscript entitled "Development and preliminary validation of a land surface image assimilation system based on the common land model" propose an image assimilation method by using the curvelet transform to denoise the observational data with only the primary structural information to be assimilated. Preliminary results showed that this assimilation method can adjust the structures of model soil moisture based on the observed spatial structure characteristics, increasing the spatial similarity of soil moisture between the model and the observation. This image assimilation method shows potential for improving the forecast of short-term climate variability related to soil moisture anomalies. However, benefit of the image assimilation is not well evaluated. That is, the paper should show more results regarding the advantages of the image assimilation over traditional single point assimilation. The paper is generally well built up. However, still this manuscript needs to be improved greatly, especially regarding the issues mentioned above. Efforts should be made to improve the readability. I think this paper can be considered for publication after some issues/questions are resolved/explained.

**Response:**

Thanks to your valuable comments and suggestions. Following your suggestions, we have revised the manuscript carefully from the beginning to the end. The point-by-point response is listed below according to your specific comments.

**Major issues:**

1. There are many other image denoising techniques, why use curvelet for land surface images? Curvelets are anisotropic, they have a high directional sensitivity and are very efficient in representing vortex edges. Therefore, the curvelet transform is suitable for geophysical fluids. But what is the argument for choosing it for land surface?

**Response:**

As the reviewer highlighted, the basis function of curvelet analysis exhibits anisotropic characteristics, thereby demonstrating its exceptional capability in accurately reproducing the rapidly-

30     evolving properties of earth fluids. Although the soil moisture does not exhibit rapid temporal variations, there are many small-scale spatial structures of soil moisture due to the high spatial heterogeneity of soil. Therefore, the curvelet analysis method is selected as a more effective approach to capture the intricate local variations of soil moisture.

    Indeed, a variety of mathematical image analysis techniques are available. For instance, the Fourier

35     decomposition method and wavelet analysis method are commonly employed in meteorological research. However, the Fourier analysis method primarily focuses on the average feature of the sequence at different frequencies, and lacks the ability to accurately describe the regional variations. The Wavelet analysis could provide more detailed variation information in the time-frequency domain, but its basis functions with isotropic characteristic limit the ability to accurately represent the characteristics of small-

40     scale spatial variations.

    On the other hand, the curvelet analysis method has been selected to fulfill the requirements of variational data assimilation. The curvelet transform is an observation operator in an image assimilation system. During the process of minimizing the cost function in variational data assimilation, the adjoint function of the observation operator becomes necessary. The adjoint function of curvelet analysis is just

45     its inverse transformation, which proves to be a highly advantageous property for minimizing the cost function in a variational data assimilation system.

2. The argument for choosing the threshold of 0.5 for the curvelet denoising is not convincing enough. Probably different thresholds will lead to different assimilation results. If it is true. How should understand this?

50     **Response:**

    Thanks for your valuable suggestions. Just as the reviewer pointed out, the image assimilation system determines the spatial structural characteristics of assimilation according to the threshold values, and different threshold values could result in certain variations in the spatial structure of assimilation.

    Naturally, a higher threshold can capture more spatial structural features of the observed variables,

55     but the presence of observation errors imposes limitations on its continuous increase. More discussions have been added to the revised manuscript in Line 366-404 to prove that a threshold of 0.5 can not only capture the spatial structure information of observation data, but also mitigate the impact of observational errors. The specific content comprises the following three aspects:

(1) The definition of the threshold σ has been further elaborated in order to provide a clearer rationale for its selection. This detailed description has been incorporated into line 370-371 of the revised manuscript. The specific wording is as follows:

The threshold σ means the modulus of the decomposition coefficient falls within the first 100*σ% percentile. For instance, a value of 0.5 indicates that the mode retaining the top 50% of decomposition coefficient.

(2) By employing the spatial correlation method, we demonstrate that a threshold of 0.5 adequately captures the primary spatial information derived from soil moisture observations, the following discussions have been added to Line 366-404 of the revised manuscript:

[revised manuscript text omitted]

lines represent threshold values of 0.4, 0.5, 0.6, 0.7 and 0.8, respectively.

The revised manuscript now includes the newly added Figure 4, which has been incorporated into

130    the section on observational error analysis along with its corresponding discussion.

And the relevant reference has been added to the revised manuscript:

Daley, R.: Atmospheric Data Analysis, Cambridge University Press, Cambridge, 1991.

3. Don't understand why there is no error covariance matrixes involved in the term J_1 in Equation (5).

**Response:**

135    As the reviewer emphasized, the data assimilation typically involves the covariance of observation

error and background error. But in this study, the image term in the cost function serves only as a weak

constraint to adjust the spatial structure of the analysis field within the image assimilation system, thereby

the error covariance is not necessary. Additionally, as explained in response to question 2, we have

elucidated in detail that the significance of setting a threshold value for effectively filtering out erroneous

140    information from the observed image.

**Minor comments:**

1. No information of the used atmospheric forcing data.

**Response:**

145    The information of used atmospheric forcing data is given in Line 166:

Atmospheric forcing conditions provide constraints on land-surface models. The quality of

atmospheric forcing data greatly affects the ability of land surface models to realistically simulate land

surface conditions. The atmospheric forcing dataset used to drive the CoLM in this study includes the

downward short-wave solar radiation at surface, downward long-wave radiation, near-surface air

150    temperature, specific humidity, precipitation rate, surface atmospheric pressure, U-component wind

speed, and V-component wind speed. It has a temporal resolution of three hours (at 0000 UTC, 0300

UTC, 0600 UTC, etc.) and the spatial resolution is T62 (about 1.875°) (Qian et al., 2006). The forcing

dataset was derived through combining observation-based analyses of monthly precipitation and surface

air temperature with intramonthly variations from the National Centers for Environmental Prediction-

155    National Center for Atmospheric Research (NCEP-NCAR) reanalysis. To correct the spurious long-term

changes and biases in the NCEP-NCAR reanalysis precipitation, surface air temperature, and solar radiation fields, Qian et al. (2006) combined the intramonthly variations from the NCEP-NCAR 6 hourly reanalysis with monthly time series derived from station records of temperature and precipitation. It is shown that the CLM3 reproduces many aspects of the long-term mean, annual cycle, interannual and

160    decadal variations when it was forced by this dataset.

We have added the relevant reference to the revised manuscript:

Qian, T., Dai, A., Trenberth, K. E., and Oleson, K. W.: Simulation of global land surface conditions from 1948 to 2004. Part I: Forcing data and evaluations, J. Hydrometeorol., 7, 953– 975, doi:10.1175/JHM540.1., 2006.

165    2. Figure 9: the correlations generally decline until the middle of July and then increase, how to understand this?

**Response:**

The occurrence of this phenomenon is attributed to the amount of precipitation in the driving data. To elucidate this matter, we superimpose the temporal variation of precipitation in the forced data within

170    the study period (indicated by gray shading) on Figure 10 of the original manuscript in Line 398.

The following discussions have been added to Line 593-600 of the revised manuscript:

It is important to note that the SCC exhibits a clear temporal variation, which does not necessarily imply a time-varying assimilation effect. This can be attributed to the dominant influence of precipitation on the changes in the SCC. Hence, Figure 13 also includes the hourly total precipitation (represented by

175    grey bars) in the model domain. The changes in precipitation exhibit a strong correlation with the SCC. From May 16 to June 15, there is minimal precipitation, corresponding to sustained high SCCs of soil moisture (red line) after assimilation. Subsequently, as precipitation increases, the SCC gradually diminishes. From August 15 to September 1, the SCC exhibits an inverse variation with decreasing precipitation.

180

[Figure]

Figure 13: Hourly variations of the spatial correlation coefficient of the surface (red and blue solid lines) and subsurface (black and gray solid lines) soil moisture between the observations and the experiments with (black and red solid lines) and without (black and gray solid lines) image assimilation, and the precipitation in the forced data was indicated by gray shading. After the vertical dashed line, it is the prediction period.

3. line 62-67: the sentences are ambiguous and hard to follow, please clarify to be concise and accurate.

**Response:**

Thanks for your valuable suggestions. We have revised these sentences to make them clear and concise.

The sentence "However, in reality, the observation quality varies sharply across regions, and the strong spatial heterogeneity of soil variables also tends to cause large spatial variations in the accuracy of surface variables simulated by the land surface model (Li, 2013; Li et al., 2020b). This leads to the regional differences in the accuracy of the estimations of observation error and background error in the single-column assimilation, and ultimately causes discontinuities in the spatial structure of the anomalies in the analyzed soil moisture fields." has been revised as follows.

Due to the non-uniform spatial distribution of precipitation, as well as the heterogeneous spatial distribution of soil properties, land cover types and topographic elevations, there are significant variations in the spatial distribution of soil moisture (Tian et al., 2021). The estimation of soil moisture by the land surface model is adversely impacted by the uncertainties in atmospheric forcing, model dynamics and parameterization, leading to significant spatial variations in the accuracy of simulated surface variables

(Li, 2013; Li et al., 2020b). Furthermore, there are regional differences in the accuracy of the estimation of the observation error and the background error resulting from the single column assimilation, which ultimately contribute to the discontinuity of the abnormal spatial structure in the analyzed soil moisture field.

And the new reference has been added to the revised manuscript:

Tian, S., Renzullo, L. J., Pipunic, R. C., Lerat, J., Sharples, W., Donnelly, C.: Satellite soil moisture data assimilation for improved operational continental water balance prediction, Hydrol. Earth Syst. Sci., 25(8): 4567-4584, https://doi.org/10.5194/hess-25-4567-2021, 2021.

4. Section 3.1: It is indicated in line 230 that resolution of the soil moisture reanalysis data is 31 km, while line 235 states "increased to 9 km".

**Response:**

Thanks for your valuable suggestion. The description of line 230 of the original manuscript has now been revised to "9 km". In line 235, "increased to 9 km" means the resolution of ERA5-land is increased from 31 km (original resolution of ERA5) to 9 km.

5. line 252: To demonstrate the benefit of the image assimilation and evaluate its advantages over traditional single point assimilation, if set J_O=0 in equation (5), authors should do one more set of experiments performing single point assimilation. Another option is to first do J(x) = J_B + J_O as conventional assimilation, and then do J(x) = J_B + J_O + J_I to see the benefit of the image assimilation.

**Response:**

Thanks to your valuable suggestion. According to your opinions, we have added a specialized section in the revised manuscript to facilitate a comparative analysis of the disparity in the effectiveness between the prevailing single point assimilation method and image assimilation method. The following analysis results have been added into the revised manuscript. Please refer to Line 419-458 for detailed information.

3.4 The influence of image assimilation constraints

Two sets of ideal experiments are designed to validate the impact of image assimilation and evaluate its superiority over traditional single column assimilation in adjusting the spatial distribution structure of

soil moisture. The ideal observational data for assimilation is the ERA5-Land reanalysis soil moisture. The first set corresponds to the conventional assimilation experiment, where $J_{(x)} = J_B + J_o$, as described by Equation (5). Another set is image assimilation experiment, where $J_o = 0$ in Equation (5), indicating that $J_{(x)} = J_B + J_I$.

The process of data assimilation entails leveraging the discrepancy between observed data and background field, in conjunction with a priori knowledge of observation error and background error, to derive an analysis field that closely approximates the true value. The primary challenge in single column assimilation lies in acquiring precise prior information regarding observation error. The spatial distribution of observation error for a specific single column assimilation experiment is illustrated in Fig. 6. In consideration of the necessity for an ideal experiment, it is assumed that the observation error outside the China region is negligible, while a significant error is presumed within the China region, so as to emphasize the impact of observation error on assimilation results.

[Figure]

Figure 6: Spatial distribution of observation errors.

The spatial distributions of soil moisture for the ideal observation data and different experiments at 0000 UTC on May 1, 2016 are given in Figure 7. The spatial distribution of surface soil moisture in ERA5-Land is illustrated in Fig. 7a. The northern Siberian region of the selected area exhibits a relatively high soil moisture content overall, with a ring-shaped distinct wet zone in the northwest. The central region stretching from Xinjiang to western Mongolia is a significant arid area. However, the soil moisture in the Tianshan Mountains is wet. The soil moisture of the Qinghai-Tibet Plateau region gradually decreases from west to east. The soil moisture in southern Qinghai, Hunan and Jiangxi is characterized by high level of saturation, while Gansu, Ningxia and Hebei experience relatively arid soil conditions. Figure 7b is the distribution of soil moisture in the control experiment (background field). It is evident that there are significant disparities in the spatial distribution of soil moisture when compared with the

reanalysis data. In the control experiment, a dry region extends from west to east in the northern area of Lake Baikal, while eastern Kazakhstan and central Inner Mongolia also exhibit arid conditions.

Figure 7c shows the results of the single point assimilation experiment. The observation error outside the China region is relatively minimal, indicating a strong correspondence between the analysis field and the observation data, and the overall distribution also exhibits a high degree of conformity with the observation. The analysis field in China region, however, closely resembles the background field. Nevertheless, there is a significant disparity between the observed soil moisture and that of the background field, indicating a lack of adjustment based on observed information.

Figure 7d is the assimilation results of the image assimilation ideal experiment. It is evident that image assimilation effectively adjusts the distribution pattern of soil moisture. The above-mentioned characteristics of moist soil moisture in the northwest region of the observation field, the arid region of Xinjiang and Mongolia, and the humid region of the Tianshan Mountains are all well reflected in the analysis field.

[Figure]

Figure 7: Spatial distributions of surface soil moisture for (a) ERA5-Land, (b) CTL experiment, (c) analysis field of conventional assimilation experiment, and (d) analysis field of image assimilation experiment at 00:00 UTC on May 1, 2016.

---

## Author Comment (AC3)

**Overall evaluation:**

This is an interesting manuscript that proposes a new image assimilation system to improve the spatial structure accuracy of soil moisture in land surface models. The method is innovative by introducing image observations and curvelet transform to optimize the model spatial patterns. The experiments generally demonstrate the capability of the proposed approach. I think this manuscript merits publication after addressing several issues.

**Response:**

Thanks to your valuable comments and suggestions. Following your suggestions, we have revised the manuscript carefully from the beginning to the end. The point-by-point response is listed below according to your specific comments.

**Major comments:**

1. The introduction needs to be improved. Soil moisture is widely assimilated in land surface models, the authors should clearly point out the limitations of current land data assimilation systems in representing spatial patterns and explain why improving spatial accuracy is important. More discussions are needed on existing studies that tried to retain spatial information in land DA. This will help highlight the motivation and significance of the current study.

More details are needed. For example, why the western East Asia is selected as the experimental region. Even in state-of-the-art land surface models, the soil hydrothermal processes (particularly over the Tibetan Plateau) are not generally well represented, it is challenge for most of the models to obtain a reliable soil moisture simulation. Therefore, is this a good choice for selecting this region as the study area?

**Response:**
The following content is added in the introduction part to emphasize the significance of spatial structure adjustment.

(1) Some discussions on the significance of spatial structural characteristics of soil moisture has been added to Line 59-68 of the revised manuscript.

Spennemann et al. (2018) emphasized the significance of the identification of Land-Atmosphere interaction region, which is crucial to enhance the weather/seasonal forecast and the better understanding

of the physical mechanisms involved. Because in these hotspot regions, soil moisture variability has the potential to modulate the atmospheric conditions by changing the latent- and sensible energy fluxes on time scales ranging from diurnal to seasonal (Seneviratne et al., 2010). Zhu et al. (2023) revealed that positive (negative) abnormal soil moisture in the eastern (western) Qinghai-Tibet Plateau during spring is associated with increased precipitation and runoff in the Yangtze River Basin during summer, while the opposite holds true. Xu et al. (2021) highlighted that the presence of extensive snow cover and soil moisture anomalies in Siberia during spring alters the thermal conditions of both the surface and atmosphere throughout summer, and then leads to anomalous atmospheric transient wave activities, which consequently stimulate and strengthen the atmospheric Rossby wave train, and ultimately result in abnormal summer precipitation patterns in South China.

The following references have been added to the reference part of the revised manuscript:

Spennemann, P.C., Salvia, M., Ruscica, R. C., Sörensson, A. A., Grings, F., Karszenbaum, H.: Land-atmosphere interaction patterns in southeastern South America using satellite products and climate models. Int J Appl Earth Obs Geoinformation, 64, 96-103, https://doi.org/10.1016/j.jag.2017.08.016, 2018.

Seneviratne, S. I., Corti, T., Davin, E. L., Hirschi, M., Jaeger, E. B., Lehner, I., Orlowsky, B., Teuling, A. J.: Investigating soil moisture–climate interactions in a changing climate: A review. Earth Sci. Rev., 99(3-4), 125-161, https://doi.org/10.1016/J.EARSCIREV.2010.02.004, 2010.

Zhu, C., Ullah, W., Wang, G., Lu, J., Li, S., Feng, A., Hagan, D., F., T., Jiang, T., Su, B.: Diagnosing potential impacts of Tibetan Plateau spring soil moisture anomalies on summer precipitation and floods in the Yangtze River Basin. J G R Atmospheres,128(8), 10.1029/2022jd037671, 2023.

Xu, B., Chen, H., Gao, C., Zeng, G., Huang, Q.: Abnormal change in spring snowmelt over Eurasia and its linkage to the East Asian summer monsoon: The hydrological effect of snow cover. Front Earth Sci, 8: 594656, 10.3389/feart.2020.594656, 2021a.

(2) Some discussions about inadequate ability of current single point assimilation method to adjust the spatial structures have been added to Line 79-89 of the revised manuscript.

Furthermore, there are regional differences in the accuracy of the estimation of the observation error and the background error resulting from the single column assimilation, which ultimately contribute to

the discontinuity of the abnormal spatial structure in the analyzed soil moisture field. The estimation of single-point observation error and background error through statistical methods is characterized by significant uncertainty, while point-by-point assimilation methods have limitations in capturing spatial information from neighboring pixels. In addition, the bias correction is commonly employed to rectify the discrepancy between model simulations and observations prior to assimilation. The prevailing assimilation system primarily addresses the bias by incorporating scale adjustments into the model simulation based on observed data. The spatial distribution structure information, however, is compromised as a result of rescaling (Zhou et al., 2019).

The following reference has been added to the reference part of the revised manuscript:

Zhou, J., Wu, Z., He, H., Wang, F., Xu, Z., Wu, X.: Regional assimilation of in situ observed soil moisture into the VIC model considering spatial variability, Hydrol. Sci. J., 64:16, 1982-1996, 10.1080/02626667.2019.1662024, 2019.

(3) The introduction of research progresses on the adjustment of spatial information of soil variables is added to Line 94-108 of the revised manuscript.

The uneven spatial distribution of precipitation and the heterogeneousness of soil properties, land cover types and topography would result in significant spatial variations in the characteristics of soil moisture (Tian et al., 2021). The effectiveness of estimating soil moisture using observational data is limited due to significant spatial heterogeneity. Therefore, a lot of studies strive to acquire the precise spatial structural information of soil moisture to the greatest extent possible. Pauwels et al. (2001) employed the nudging technique to incorporate spatial structure information derived from remote sensing soil moisture observations, and obtained enhanced predictions of runoff. Han et al. (2012) examined the constraints of introducing the horizontal correlation features of satellite soil moisture observation data during land surface data assimilation. The findings demonstrated that incorporating surrounding observations and spatial horizontal correlation structure information may improve the analysis field of soil moisture in uncovered grids. The regional soil water assimilation scheme developed by Zhou et al. (2019) incorporates an empirical approach and accounts for spatial variability, resulting in significantly improved accuracy of soil moisture simulation in both temporal and spatial dimensions. The findings of these studies suggest that enhancing soil moisture levels is of utmost importance; however, it is equally

crucial to acquire a more precise comprehension of the spatial distribution of soil moisture for effective management strategies, particularly in key regions like the Qinghai-Tibet Plateau where land-air interactions are significant and there are large spatial variations of soil moisture.

90     The following references have been added to the reference part of the revised manuscript:

Tian, S., Renzullo, L. J., Pipunic, R. C., Lerat, J., Sharples, W., Donnelly, C.: Satellite soil moisture data assimilation for improved operational continental water balance prediction, Hydrol. Earth Syst. Sci., 25(8): 4567-4584, https://doi.org/10.5194/hess-25-4567-2021, 2021.

Pauwels, V. R. N., Hoeben, R., Verhoest, N. E. C., Troch, F. P. D.: The importance of the spatial

95     patterns of remotely sensed soil moisture in the improvement of discharge predictions for small-scale basins through data assimilation, J Hydrol, 251(1-2), 88-102, https://doi.org/10.1016/S0022-1694(01)00440-1, 2001.

Han, X., Li, X., Hendricks Franssen H. J., Vereecken, H., Montzka, C.: Spatial horizontal correlation characteristics in the land data assimilation of soil moisture. Hydrol. Earth Syst. Sci., 16(5), 1349-1363,

100     https://doi.org/10.5194/hess-16-1349-2012, 2012.

Zhou, J., Wu, Z., He, H., Wang, F., Xu, Z., Wu, X.: Regional assimilation of in situ observed soil moisture into the VIC model considering spatial variability, Hydrol. Sci. J., 64:16, 1982-1996, 10.1080/02626667.2019.1662024, 2019.

The following contents are added to explain the purpose of research area selection in Line 129-132:

105     The study area selected in this research is mainly East Asia, encompassing the alpine regions of Siberia, the vegetative regions of eastern China, as well as the Qinghai-Tibet Plateau and desert regions of western China. The estimation of observation error and model error becomes more challenging in the Tibetan Plateau region, particularly for single point assimilation. Including the plateau region can effectively showcase the advantages of image assimilation method.

110     2、The authors claim the capability of improving deep soil moisture through assimilating surface data. But no clear explanations are given on the underlying mechanism. Some discussions should be added regarding how the surface information propagates to deeper layers through model physics. Moreover, when comes to the complexity of the soil hydrothermal processes of the Tibetan Plateau (e.g., soil freezing and thawing) and the difficult of the model to parameterize these processes, how to

115     Response:

According to the comments of the reviewer, we attempted to explain how assimilating of surface soil moisture improves deeper soil moisture on the basis of the physical process of vertical soil water movement. The actual model results were utilized to validate the gradual impact of analysis increment on deeper soil moisture. The following additional figures and associated descriptions have been added to lines 155-158 of the revised manuscript:

[revised manuscript text omitted]

3、The soil moisture product of EAR5_Land was assimilated and used to assessment, please explain in detail the rational for this approach. Why not choose an independent soil moisture product to evaluate the assimilation results?

**Response:**

Thanks for your valuable suggestion. The CLDAS reanalysis data is chosen as the independent dataset, and an additional verification analysis of the assimilation results based on the CLDAS data is conducted. The CLDAS product is produced by using the near-real-time CLDAS atmospheric drive product, which incorporates a larger amount of ground station observation data and higher quality background field, the dataset exhibits excellent quality and offers high spatio-temporal resolution data in the China region (Shi et al., 2011; Liu et al., 2019).

The following description of CLDAS reanalysis data is added to Line 289-297 Of the revised manuscript:

The soil volume water content reanalysis product V2.0, generated by the land surface data assimilation system CLDAS of the National Meteorological Information Center of China Meteorological Administration, covers the Asian region (0–65°N, 60–160°E). The temporal resolution is 1 hour, and the spatial resolution is 0.0625°. The vertical direction is divided into five layers: 0–5 cm, 5–10 cm, 10–40 cm, 40–100 cm, and 100–200 cm. The CLDAS product is produced by using the near-real-time CLDAS atmospheric drive product, which incorporates a larger amount of ground station observation data and higher quality background field to drive various land surface models (such as CLM 3.5, CoLM and Noah-MP). As a result, the dataset exhibits excellent quality and offers high spatio-temporal resolution data in the China region (Shi et al., 2011; Liu et al., 2019). The CLDAS reanalysis data is therefore chosen as the independent dataset, and an additional verification analysis of the assimilation results based on the CLDAS data is conducted.

The following evaluation results are added to Line 402-??? of the revised manuscript:

The overlapping region (22–50°N,73–117°E) between the CLDAS data and the model region is selected for analysis. The spatial correlation coefficients of soil moisture before and after assimilation to CLDAS data are also computed, aiming to quantitatively assess the accuracy of the adjustment in soil

moisture spatial distribution structure by the image assimilation system. The image assimilation results in a notable increase in the spatial correlation coefficient between CoLM soil moisture and the first layer (0–5cm) soil moisture of CLDAS, as depicted in Fig. 14a. Throughout the assimilation and prediction stages, this correlation coefficient consistently surpasses that of the CTL experiment, with a maximum

220 value of 0.79. Moreover, after assimilation, there is an average increase in spatial correlation coefficient from 0.67 to 0.71. The image assimilation brings a more significant increase in the spatial correlation coefficient of soil moisture in the second layer (5–10 cm), as depicted in Fig. 14b. The highest spatial correlation coefficient reaches 0.79, while the average value increases from 0.67 to 0.73. The verification results of independent data further confirm that the image assimilation system has a strong capability in

225 adjusting the spatial structure of soil moisture, particularly in relation to subsurface soil moisture.

[Figure]

Figure 14: The spatial correlation coefficients of the CLDAS products to the CTL experiment (red solid line) and image assimilation experiment (blue solid line) for the first layer (0–5cm) and second layer (5–10 cm) from May 16 to September 30, 2016.

The following references have been added to the reference part of the revised manuscript:

Shi, C., Xie, Z., Qian, H., Liang, M., and Yang, X.: China land soil moisture EnKF data assimilation based on satellite remote sensing data, Sci China Earth Sci, 41, 375-385, https://doi.org/10.1007/s11430-010-4160-3, 2011.

Liu, J. G., Shi, C. X., Sun, S., Liang, J. J., Yang, Z.-L.: Improving land surface hydrological simulations in China using CLDAS meteorological forcing data. J Meteorol Res-prc, 33(6), 1194-1206, https://doi.org/10.1007/s13351-019-9067-0, 2011.

4. The evaluation of the image assimilation system relies heavily on the EAR5_Land soil moisture reanalysis data, in-situ soil moisture observations from dense observation network are recommended to be used in this study.

**Response:**

Thanks to your valuable suggestion.

In order to enhance the reliability of the results, we incorporate the CLDAS reanalysis data as an independent verification dataset. The CLDAS reanalysis data assimilates a substantial amount of high-density ground observation data in China. We have reasonable grounds to believe that the verification outcomes based on the CLDAS reanalysis data exhibit similarity with those derived from in situ observations. Numerous studies have also demonstrated that the CLDAS reanalysis data bear a strong resemblance to actual site observation data, as evidenced by a national regional average correlation coefficient of 0.89, a root-mean-square error of 0.02 m3/m3, and a deviation of 0.01 m3/m3. So the CLDAS and ERA5-Land datasets are chosen for separate assessment of their assimilation effects (Shi et al., 2011; Liu et al., 2019).

5. The conclusion needs to be strengthened by summarizing key findings, pointing out limitations and discussing future outlooks. Comparisons with existing studies are needed to highlight the specific improvements.

**Response:**

Thanks for your valuable suggestion. The conclusion and discussion part has been further refined, encompassing a recapitulation of significant findings for the study, and an emphasis on the limitations and future prospects of image assimilation methods, as well as the inclusion of comparative analysis with existing studies. The revised version of the conclusion in Line 662-705 is presented below.

[revised manuscript text omitted]

**Minor issues:**

1、Line 234, the word "ECM-WF/IFS" should be "ECMWF/IFS"

**Response:**

Thanks for your valuable suggestion. We have revised the word "ECM-WF/IFS" to "ECMWF/IFS" in Line 234.

2、Line 271, "seprately" should be "separately".

**Response:**

Thanks for your valuable suggestion. We have revised the word "seprately" to "separately" in Line 271.

---

## Author Comment (AC4)

**Overall evaluation:**

The manuscript proposed a land surface image assimilation system capable of optimizing the spatial structure of the background field, and the ERA5-Land soil moisture reanalysis data was used as ideal observation to validate the assimilation system. The results of ideal experiments showed that the proposed image assimilation system exhibits a remarkable ability to adjust the spatial structure of soil moisture in a land surface model, considerably improving the prediction skill. This is an interesting study. However, the manuscript is not written well and lacks in-depth analysis. For example, in the section "Discussion and conclusions", it would be useful to add more information comparing the proposed assimilation system with existing assimilation systems.

**Response:**

Thanks to your valuable comments and suggestions. Following your suggestions, we have made more deep analysis and revised the manuscript carefully from the beginning to the end.

(1) The conclusion and discussion part has been further refined, encompassing a recapitulation of significant findings for the study, and an emphasis on the limitations and future prospects of image assimilation methods, as well as the inclusion of comparative analysis with existing studies. The revised version of the conclusion in Line 662-705 is presented below:

[revised manuscript text omitted]

70    subsequent research.

(2) In addition, the following experiments and analysis for choosing the threshold of 0.5 for the curvelet denoising have been incorporated to enhance the depth of analysis in the manuscript.

The image assimilation system determines the spatial structural characteristics of assimilation according to the threshold values, and different threshold values could result in certain variations in the

75    spatial structure of assimilation. Naturally, a higher threshold can capture more spatial structural features of the observed variables, but the presence of observation errors imposes limitations on its continuous increase. More discussions have been added to the revised manuscript in Line 366-404 to prove that a threshold of 0.5 can not only capture the spatial structure information of observation data, but also mitigate the impact of observational errors. The specific content comprises the following three

80    aspects:

A. The definition of the threshold σ has been further elaborated in order to provide a clearer rationale for its selection. This detailed description has been incorporated into line 370-371 of the revised manuscript. The specific wording is as follows:

[revised manuscript text omitted]

The revised manuscript now includes the newly added Figure 4, which has been incorporated into the section on observational error analysis along with its corresponding discussion.

And the relevant reference has been added to the revised manuscript:

Daley, R.: Atmospheric Data Analysis, Cambridge University Press, Cambridge, 1991.

(3) We have added a specialized section in the revised manuscript to facilitate a comparative analysis of the disparity in the effectiveness between the prevailing single point assimilation method and image assimilation method.

3.4 The influence of image assimilation constraints

Two sets of ideal experiments are designed to validate the impact of image assimilation and evaluate its superiority over traditional single column assimilation in adjusting the spatial distribution structure of soil moisture. The ideal observational data for assimilation is the ERA5-Land reanalysis soil moisture. The first set corresponds to the conventional assimilation experiment, where $J_{(x)} = J_B + J_o$, as described by Equation (5). Another set is image assimilation experiment, where $J_o = 0$ in Equation (5), indicating that $J_{(x)} = J_B + J_I$.

The process of data assimilation entails leveraging the discrepancy between observed data and background field, in conjunction with a priori knowledge of observation error and background error, to derive an analysis field that closely approximates the true value. The primary challenge in single column assimilation lies in acquiring precise prior information regarding observation error. The spatial distribution of observation error for a specific single column assimilation experiment is illustrated in Fig. 6. In consideration of the necessity for an ideal experiment, it is assumed that the observation error outside the China region is negligible, while a significant error is presumed within the China region, so as to emphasize the impact of observation error on assimilation results.

[Figure]

Figure 6: Spatial distribution of observation errors.

The spatial distributions of soil moisture for the ideal observation data and different experiments at 0000 UTC on May 1, 2016 are given in Figure 7. The spatial distribution of surface soil moisture in ERA5-Land is illustrated in Fig. 7a. The northern Siberian region of the selected area exhibits a relatively high soil moisture content overall, with a ring-shaped distinct wet zone in the northwest. The central region stretching from Xinjiang to western Mongolia is a significant arid area. However, the soil moisture in the Tianshan Mountains is wet. The soil moisture of the Qinghai-Tibet Plateau region gradually decreases from west to east. The soil moisture in southern Qinghai, Hunan and Jiangxi is characterized by high level of saturation, while Gansu, Ningxia and Hebei experience relatively arid soil conditions. Figure 7b is the distribution of soil moisture in the control experiment (background field). It is evident that there are significant disparities in the spatial distribution of soil moisture when compared with the reanalysis data. In the control experiment, a dry region extends from west to east in the northern area of Lake Baikal, while eastern Kazakhstan and central Inner Mongolia also exhibit arid conditions.

Figure 7c shows the results of the single point assimilation experiment. The observation error outside the China region is relatively minimal, indicating a strong correspondence between the analysis field and the observation data, and the overall distribution also exhibits a high degree of conformity with the observation. The analysis field in China region, however, closely resembles the background field. Nevertheless, there is a significant disparity between the observed soil moisture and that of the background field, indicating a lack of adjustment based on observed information.

Figure 7d is the assimilation results of the image assimilation ideal experiment. It is evident that image assimilation effectively adjusts the distribution pattern of soil moisture. The above-mentioned characteristics of moist soil moisture in the northwest region of the observation field, the arid region of Xinjiang and Mongolia, and the humid region of the Tianshan Mountains are all well reflected in the analysis field.

[Figure]

Figure 7: Spatial distributions of surface soil moisture for (a) ERA5-Land, (b) CTL experiment, (c) analysis field of conventional assimilation experiment, and (d) analysis field of image assimilation experiment at 00:00 UTC on May 1, 2016.

The analysis results mentioned above have been added into the revised manuscript. Please refer to Line 236 for detailed information.

(4) We attempted to explain the underlying physical mechanisms how the surface soil moisture assimilation improves deep soil moisture, which are elucidated through a vertical-time profile of soil moisture. The following additional figures and associated descriptions have been added to lines 155-158 of the revised manuscript:

[revised manuscript text omitted]

280 tree (orange dots).

And the point-by-point response is listed below according to your specific comments.

Minor comments:

Line 18: Please use the official name: ERA5-Land.

285 **Response:**

Thank you for your valuable suggestion. We have revised the word "ERA5_Land" to "ERA5-Land" throughout the manuscript

Line 113: "five snow layers" -> "a maximum of five snow layers".

**Response:**

290     Thank you for your valuable suggestion. We have revised the words "five snow layers" in Line 154 to "a maximum of five snow layers".

Line 125: Please add reference to the atmospheric forcing data.

**Response:**

       Thank you for your valuable suggestion. We have added the reference and relevant information
295    about the forcing data in Line 170-176 as follows.

       The forcing dataset was derived through combining observation-based analyses of monthly precipitation and surface air temperature with intramonthly variations from the National Centers for Environmental Prediction-National Center for Atmospheric Research (NCEP-NCAR) reanalysis. To correct the spurious long-term changes and biases in the NCEP-NCAR reanalysis precipitation, surface
300    air temperature, and solar radiation fields, Qian et al. (2006) combined the intramonthly variations from the NCEP-NCAR 6 hourly reanalysis with monthly time series derived from station records of temperature and precipitation. It is shown that the CLM3 reproduces many aspects of the long-term mean, annual cycle, interannual and decadal variations when it was forced by this dataset.

       The reference which has been added to the revised paper is as follows :

305    Qian, T., Dai, A., Trenberth, K. E., and Oleson, K. W.: Simulation of global land surface conditions from 1948 to 2004. Part I: Forcing data and evaluations, J. Hydrometeorol., 7, 953– 975, doi:10.1175/JHM540.1., 2006.

Line 209: Please provide the source of the raw soil moisture image plotted in Figure 1a.

**Response:**

310    Thank you for your valuable suggestion. Figure 1a shows the spatial distribution of soil moisture simulated by the CoLM land surface model on May 1, 2016. The aforementioned note has been incorporated into the revised manuscript at Line 260.

Line 266: "The depths of the top three soil layers …" => "The node depths of the top three soil layers …"

315

**Response:**

Thank you for your valuable suggestion. We have revised "The depths of the top three soil layers …" to "The node depths of the top three soil layers …".

Line 369: It is not appropriate to refer to soil at the depth of 7-28cm as deep soil.

**Response:**

320

Thank you for your valuable suggestion. We have revised "deep soil" to "The soil moisture content at a depth of 7–28 cm."

Line 445: This is not the reason to develop a new assimilation scheme for land surface model.

**Response:**

Thank you for your valuable suggestion, and we have revised the sentence to make the reason

325

more clearer.

The original sentences have been changed as follows.

The exchange of heat and water vapor between the land surface and the atmosphere plays a crucial role in influencing weather and climate change. The impact of soil moisture on atmospheric changes is frequently manifested through the persistent influence of large-scale soil moisture anomalies. The

330

construction of an assimilation system with image assimilation capability is aimed at enhancing the spatial structure accuracy of soil moisture anomalies in the initial field of land surface models.

Line 466: Please rephrase this sentence.

**Response:**

335

Thank you for your valuable suggestion. The sentence has been revised as follows.

The image assimilation system developed by this study could effectively optimize the spatial structure of soil variables in the background by incorporating constraint conditions of the observed spatial structures. The method demonstrates excellent applicability to various soil variables, effectively

mitigating the negative impact of strong spatial heterogeneity of soil on data assimilation. The key challenge in image assimilation lies in obtaining accurate spatial structure observation of soil variables. The data of ground automatic stations with high spatial-temporal resolution established in China, along with satellite observation data that can overcome natural constraints and achieve large-scale uniform observation in various terrains, are capable of providing observational images depicting the spatial structure of land surface variables for image assimilation. The effective assimilation of the spatial structural characteristics of those high-density meteorological observation data, is the primary focus of our subsequent research. However, how to establish the direct spatial structure relationship between satellite-observed brightness temperature data and soil variables, and how to repair these non-uniform data into uniformly distributed data, these are the key technical problems that need to be solved in the future.

Additionally, it should be noted that the image assimilation method and the prevailing single-point land assimilation method in current practice are not mutually exclusive. The single-point land assimilation method is more suitable for assimilating sparse observation data in key areas. However, if the image assimilation method is used to optimize the fine structure of soil moisture in specific areas, the threshold $\sigma$ mentioned above needs to be further increased, but this approach is susceptible to introducing additional observational errors. Therefore, by integrating the capacity of the image assimilation method in adjusting the large-scale spatial structure of soil variables and the capability of single-point land assimilation method in finely optimizing soil variables in crucial regions, and by leveraging the advantages offered by diverse types of meteorological observation data, we can attain more refined initial conditions for land models, which constitutes the primary objective of our subsequent research.

---

## Referee Report (RR1)

**Overall evaluation:**

Most of my concerns are well addressed in the revised manuscript. However, part of the new added content is verbose, and therefore the language needs to be improved greatly. I think this paper can be considered for publication after the following issues are resolved.

**Specific issues:**

[1]   Line 59, what is the "Land-Atmosphere interaction region"? Land-Atmosphere interactions happen everywhere on Earth, so how to identify this process? Based on my understanding, I think it should be "Land-Atmosphere interaction hotspot region".

[2]   Line 65, "while the opposite holds true" maybe not correctly used here.

[3]   The new added content (Line 59-68, 94-108) is very lengthy, please modify the sentences to make the description concise.

[4]   Line 167, CLM3 should be "CLM 3.0".

[5]   Line 278, the full name of CLDAS should be given when it appears for the first time.

---

## Author Response (AR2)

**Overall evaluation:**

Most of my concerns are well addressed in the revised manuscript. However, part of the new added content is verbose, and therefore the language needs to be improved greatly. I think this paper can be considered for publication after the following issues are resolved.

**Response:**

Thanks for your suggestions. We have modified the verbose sentences to make the description concise according to your suggestions. We also rectified the typographical errors and added the full name of CLDAS when it appears for the first time.

**Specific issues:**

[1] Line 59, what is the "Land-Atmosphere interaction region"? Land-Atmosphere interactions happen everywhere on Earth, so how to identify this process? Based on my understanding, I think it should be "Land-Atmosphere interaction hotspot region".

**Response:**

Thanks for your suggestions. We have revised the description to "Land-Atmosphere interaction hotspot region".

[2] Line 65, "while the opposite holds true" maybe not correctly used here.

**Response:**

Thanks for your suggestions. Following the third suggestion, we have concise the relevant description to "The significant influence of soil moisture on local precipitation has been extensively studied, revealing regional variations in the underlying mechanisms. Additionally, soil moisture can also trigger the atmospheric teleconnection wave trains or inducing large-scale circulation anomalies through impacting surface energy balance, which subsequently manifest as non-local and large-scale climate effects." In line 68-73.

[3] The new added content (Line 59-68, 94-108) is very lengthy, please modify the sentences to make the description concise.

**Response:**

Thanks for your valuable comments and suggestions. Following your suggestions, we have revised the content in line 59-68 of the manuscript from "Spennemann et al. (2018) emphasized the significance of the identification of Land-Atmosphere interaction region, … strengthen the atmospheric Rossby wave train, and ultimately result in abnormal summer precipitation patterns in South China." to

"The significant influence of soil moisture on local precipitation has been extensively studied, revealing regional variations in the underlying mechanisms (Douville et al., 2001; Cioni and Hohenegger, 2017). Additionally, soil moisture can also trigger the atmospheric teleconnection wave trains or inducing large-scale circulation anomalies through impacting surface energy balance, which subsequently manifest as non-local and large-scale climate effects (Gao et al., 2020)." In line 68-73.

The relevant reference is:

Douville, H., Chauvin, F., Broqua, H.: Influence of soil moisture on the Asian and African monsoons. Part I: Mean monsoon and daily precipitation, J. Climate, 14, 2381-2403, https://doi.org/10.1175/1520-0442(2001)014%3C2381:IOSMOT%3E2.0.CO;2, 2001.

Cioni, G., Hohenegger, C.: Effect of soil moisture on diurnal convection and precipitation in large-eddy simulations, J Hydrometeorol, 18, 1885-1903, https://doi.org/10.1175/JHM-D-16-0241.1, 2017.

Gao, C., G. Li, H. Che, and H. Yan.: Interdecadal change in the effect of spring soil moisture over the Indo-China Peninsula on the following summer precipitation over the Yangtze River basin, J. Climate, 33, 7063-7082, https://doi.org/10.1175/JCLI-D-19-0754.1, 2020.

And the content of line 94-108 have been revised from "Therefore, a lot of studies strive to acquire the precise spatial structural information of soil moisture to the greatest extent possible. Pauwels et al. (2001) … particularly in key regions like the Qinghai-Tibet Plateau where land-air interactions are significant and there are large spatial variations of soil moisture." to

"Therefore, a lot of studies strive to incorporate spatial structure information from soil moisture observation to land data assimilation, to enhance the accuracy of spatial pattern of soil moisture to the greatest extent possible (Pauwels et al., 2001; Han et al., 2012; Zhou et al., 2019). Enhancing soil moisture levels is of utmost importance; however, it is equally crucial to acquire a more precise comprehension of the spatial distribution of soil moisture for effective management strategies, particularly in key regions like the Qinghai-Tibet Plateau where land-air interactions are significant and there are large spatial variations of soil moisture." in line 108-114.

[4] Line 167, CLM3 should be "CLM 3.0".

**Response:**

We have revised "CLM3" to "CLM 3.0" in line 167.

[5] Line 278, the full name of CLDAS should be given when it appears for the first time.

**Response:**

We have added the full name of CLDAS (CMA Land Data Assimilation System) in line 278.